# Pre-pro is a fast pre-processor for single-particle cryo-EM by enhancing 2D classification

Szu-Chi Chung [1], Hsin-Hung Lin[2], Po-Yao Niu[1], Shih-Hsin Huang[2], I-Ping Tu [1✉] & Wei-Hau Chang [2✉]

2D classification plays a pivotal role in analyzing single particle cryo-electron microscopy images. Here, we introduce a simple and loss-less pre-processor that incorporates a fast dimension-reduction (2SDR) de-noiser to enhance 2D classification. By implementing this 2SDR pre-processor prior to a representative classification algorithm like RELION and ISAC, we compare the performances with and without the pre-processor. Tests on multiple cryo-EM experimental datasets show the pre-processor can make classification faster, improve yield of good particles and increase the number of class-average images to generate better initial models. Testing on the nanodisc-embedded TRPV1 dataset with high heterogeneity using a 3D reconstruction workflow with an initial model from class-average images high-lights the pre-processor improves the final resolution to 2.82 Å, close to 0.9 Nyquist. Those findings and analyses suggest the 2SDR pre-processor, of minimal cost, is widely applicable for boosting 2D classification, while its generalization to accommodate neural network de-noisers is envisioned.

[1] Institute of Statistical Science, Academia Sinica, 128 Academia Road, Section 2, Nankang, Taipei 11529, Taiwan. [2] Institute of Chemistry, Academia Sinica, 128 Academia Road, Section 2, Nankang, Taipei 11529, Taiwan. ✉email: iping@stat.sinica.edu.tw; weihau@gate.sinica.edu.tw

Cryo-EM (cryo-electron microscopy) uses an electron beam transmitted through a biological sample to generate projection images. The projection images of a sample can be used to reconstruct the 3D structure when many views are available[1]. For a sample of protein solution frozen in vitreous ice[2], each particle can assume arbitrary orientation that the projection images from different particles may represent different views of a 3D structure. Since cryo-EM only uses a small number of electrons for imaging to alleviate radiation damage on biological specimens, the recorded images are heavily contaminated by shot noise. To process those noisy particle images, a step-wise computation pipeline that aims to obtain a reliable 3D map of the target macro-molecule has been constructed (Fig. 1a and Fig. 1 in ref. [3]). 2D classification serves a pivotal role in the entire workflow—it curates a dataset by grouping together the particles of similar view to enhance the signal-to-noise ratio (SNR) and meanwhile discarding invalid particles or contaminants. The class averages can be used for assessing the degree of heterogeneity in data whereas the good ones are chosen for calculating an initial model. As particle images of similar orientation are related to each other by image translation and rotation, clustering alike particles entails the images to be properly aligned first. Since aligning low-SNR images is error-prone, 2D classification is a fundamentally demanding task while the results are often non-ideal. A typical 2D classification algorithm therefore couples clustering with image alignment and uses iterations to strive for the best alignment parameters and classification indices. In the era of cryo-EM "resolution revolution"[4], the computation burden of 2D classification is further aggravated by the rapid increase in the number and the size of images. A standard computation framework for 2D classification has been established since the early development of single-particle cryo-EM[5]—this framework combines K-means clustering with a multi-reference alignment

(MRA) approach where a number of images are chosen from the data to serve as initialization seeds and alignment references. To mitigate the issue by initialization, RELION classification[6]—a now widely used classification method, employs maximum-likelihood (ML) approach[7] to do MRA, allowing each image to be compared with all images in all possible rotations and translations. An image is then allocated to all classes, yet with different probabilities derived by maximizing the likelihood of observing the experimental dataset using the expectation-maximization algorithm[8]. This originally slow process has been recently accelerated thanks to GPU parallelism[9]. As a result, RELION has become a popular approach. Nonetheless, as RELION reports all classes—clear and blurred ones, human inspection is required to select good classes. Some of the good classes can still be heterogeneous as they have the potential to attract less frequent views or low-SNR images[3,10]. Moreover, optimal outcome of RELION may depend on customer-specified regularization parameters or a good guess on the number of classes. Currently, the best classification results can be obtained from ISAC—iterative stable alignment and clustering[11]. ISAC uses repeated stability tests to validate the members of each class to ensure its homogeneity. In addition, ISAC restricts the size of each class with the same bound by using a modified K-means to suppress the above-mentioned attractor effect[3,10]. These features make ISAC an attractive approach when one works on a very heterogeneous dataset. Since ISAC automatically discards the classes that are not stable or reproducible, it may not need human intervention when it comes to selecting good classes. However, ISAC is recommended only for tough problems because it is extremely time consuming.

Here, instead of inventing a new 2D classification algorithm, we propose a pre-processing strategy to enhance the performance of existing algorithms. The rationale comes from a finding that salient features of cryo-EM particles can emerge from the

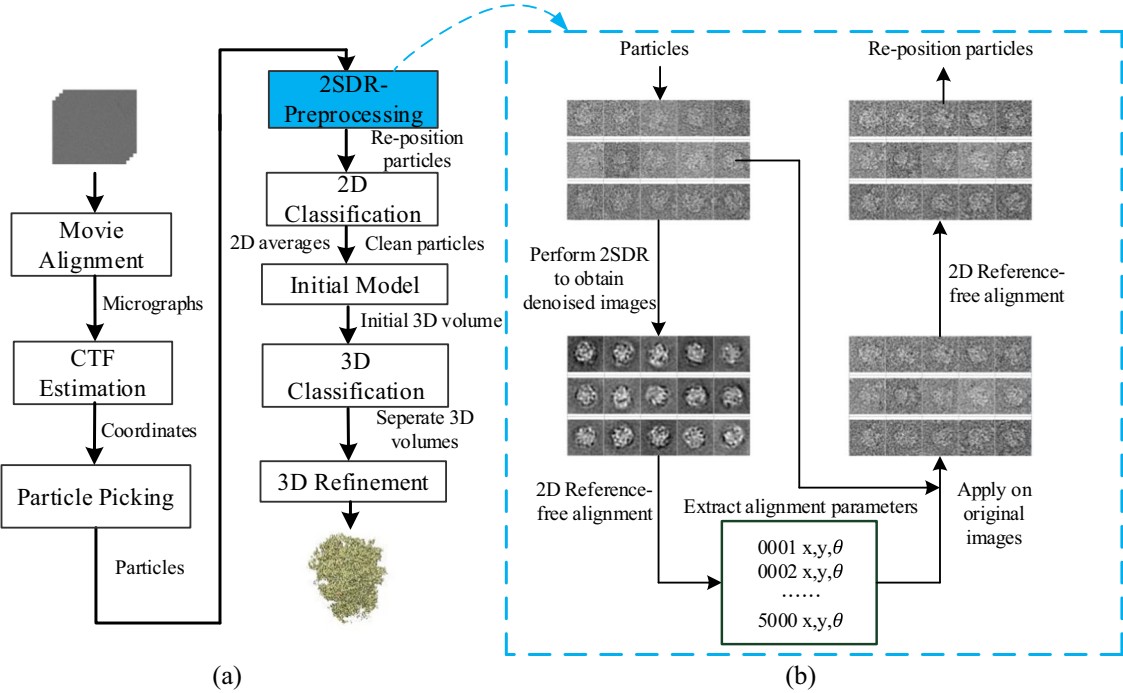

**Fig. 1 The flow charts of the processing and the pre-processing. (a)** A single Cryo-EM image processing workflow. (**b**) The workflow of proposed pre-processing. The upper panel in the left column represents the original particle images; The lower panel in the left column represents the denoised version. The bottom panel in the central column shows the *x*-and-*y* shifts and in-plane rotation angle reported by a reference-free alignment procedure applied on the denoised particles. The lower panel in the right column represents the re-positioned particle images obtained by applying the alignment parameters in the bottom central panel to the original particle images in the upper left panel. The upper right panel represents the original particles further fine-tuned by applying additional rounds of reference-free alignment to the lower right particles.

surrounding through denoising by a fast dimension-reduction method (2SDR)[12] (Fig. 1(b)). We envision that these denoised particles may give better estimates of the parameters of alignment (see "Methods").

By installing this 2SDR pre-processing prior to a concurrent 2D classification algorithm such as RELION or ISAC, we compare the performances of classification with and without the pre-processing. For the classification experiments, we use various experimental cryo-EM particle datasets, including two small datasets, 70S ribosome ($N = 5000$) and beta-galactosidase ($N = 5672$), and four large sets: 80S ribosome ($N = 105,247$), TRPV1 ion channel ($N = 35,645$ for the curated set, $N > 80,000$ for the raw dataset), and nanodisc-embedded TRPV1 ($N = 218,805$). The tests demonstrate that the pre-processing demands minimum computation cost—it consumes less than an hour even for the large dataset of 80S ribosome, and collectively report that the pre-processing can give in return increased yield of particles, increased number of good classes and appreciable reduction of the time or iterations spent on classification. Notably, reprocessing the large datasets with the aid of the pre-processing has resulted in improved 3D maps concomitant with measurable advances in resolution to surpass those in the previous reports, in particular in the cases of more heterogeneous data. In summary, we demonstrate that the 2SDR pre-processing is a cost-effective approach to boosting the performance of state-of-the-art 2D classification algorithms to impact the outcome of 3D reconstruction.

## Results

**Pilot test on 70S ribosome shows the benefits of denoising**. We first tested the effect of denoising on the alignment and grouping of particle images. Since we found direct usage of denoised particles in 2D classification could not generate reliable results, we propose a strategy based on the following heuristics for utilizing the denoised information. In this experiment, we used the 2SDR method to generate the denoised surrogates of the 5000 70S ribosome particles. From the set of denoised particles, we randomly picked five as the reference particles. For each reference in Supplementary Fig. 1 (column (a)), we searched in the denoised set to find the twenty most resembling particles by using FRM2D algorithm[13]. The best alignment parameters of rotation angles and translational shifts in x-and-y-direction were recorded for each particle, but applied to the original non-denoised particles to generate an aligned average. As shown in Supplementary Fig. 1 (column (c)), these averages resemble the projections of 70S ribosome (column (d)) and display more details than those obtained from the control experiment without denoising (column (b)). Deeper investigation reveals that the particle set found using denoising does not overlap well with that found without denoising. We designed a simulation study allowing for measuring the occurrence of true positives. With simulated noisy images of 70S ribosome (SNR = 0.01, defocus range 1.5–2.0 μm) prepared as described in Supplementary, we found the frequencies of true positive were 11.2% and 3.5% for with and without the denoising, respectively (Supplementary Fig. 2). With the SNR increased to 0.05, the frequencies are increased to 94.4% and 61.5% respectively (Supplementary Fig. 3). We further tested the effect of defocus—with SNR kept the same (0.05) but the defocus lowered to 1.0–1.5 μm, the frequencies drop to 93.7% and 51.3% (Supplementary Fig. 4). These results show that the chances of grouping identical or similar particles are markedly higher when denoising is used where the gain by 2SDR is more pronounced when the SNR is lower or the defocus is smaller.

**Pre-Pro can be coupled with RELION to give better results**. To test whether or not this pre-processing can be coupled with a 2D classification algorithm, we used two small experimental datasets, the 70S ribosome as used for the pilot experiment and the beta-galactosidase (see "Methods"). We compared the result from feeding the images re-positioned by the pre-processing to a classification algorithm with that from using the original images of no pre-processing. To evaluate the performance of a classification process, we used three performance indices: the number of good classes, the resulting initial model from class averages, and the time spent on classification.

We began our test with RELION since it has been used to solve the largest number of cryo-EM structures deposited into the PDB databank. When the pre-processing is used prior to RELION, we abbreviate the procedure as P-RELION. To do so with small datasets, we used RELION (2.0) and prescribed the number of classes to be 50 for these two small sets while leaving the remaining parameters unchanged as the default values. The class averages were sorted according to a quality index of "rlnClassDistribution", which places the most populated classes on the top rows (Fig. 2). Usually, these classes coincide with the classes of topmost quality. As a result, evaluation of class quality could be performed by inspection from the top row. As shown in Fig. 2(b), clear classes of 70S ribosome obtained through P-RELION spills over into the third row. By contrast, without the pre-processing, there are still blurred classes in the second row (Fig. 2(a)). These alleged improvements are consistent with the statistics that report the accuracy in translations and rotations (see Supplementary Fig. 5). As a result, the number of clear classes is increased by 30% for 70S ribosome and 40% for beta-galactosidase. The yield of resulting particles is increased only marginally, from 90% to 94% for 70S ribosome and from 78% to 84% for beta-galactosidase. These findings together suggest, with the aid of the pre-processor, the homogeneity of each class would be potentially improved when similar number of particles are dispersed into more classes. We further used the class averages to calculate the initial model of beta-galactosidase, by which we used PRIME[14] for its speed and robustness. Strikingly, in the absence of symmetry constraint, the 3D model from good classes generated by P-RELION (Fig. 2(f)) displays the symmetry character of beta-galactosidase while the class averages from RELION without the pre-processing does not (Fig. 2(e)). Importantly, the model from P-RELION matches better with the golden 3D model, as judged by the initial model to golden model FSC (Fig. 2(g)) and by docking of an atomic model of beta-galactosidase (Supplementary Fig. 6). These tests demonstrate the pre-processing can be successfully coupled with RELION to improve the classification results to yield better initial model. We also investigated RELION (3.0)[15], which is 1.6X faster than RELION (2.0), to have similar findings.

Though RELION classification has been accelerated by GPU parallelism, we are still curious about if there would be any measurable impact on the kinetics of classification introduced by the pre-processing. The overall time, as documented in Table 1, remains roughly the same with the pre-processing because it is set by the default number of iterations, which is 25. To explore whether or not the pre-processor could further accelerate RELION 2D classification, we examined the evolution of particle yield and initial model of the beta-galactosidase produced from different iterations. As shown in Supplementary Fig. 7, P-RELION with 10 iterations can result in a yield of particles and and an initial model comparable to those from RELION with 25 iterations.

**Pre-Pro makes ISAC faster and save closer-to-focus particles**. To test ISAC, we set the size limit of each class to 100 for both

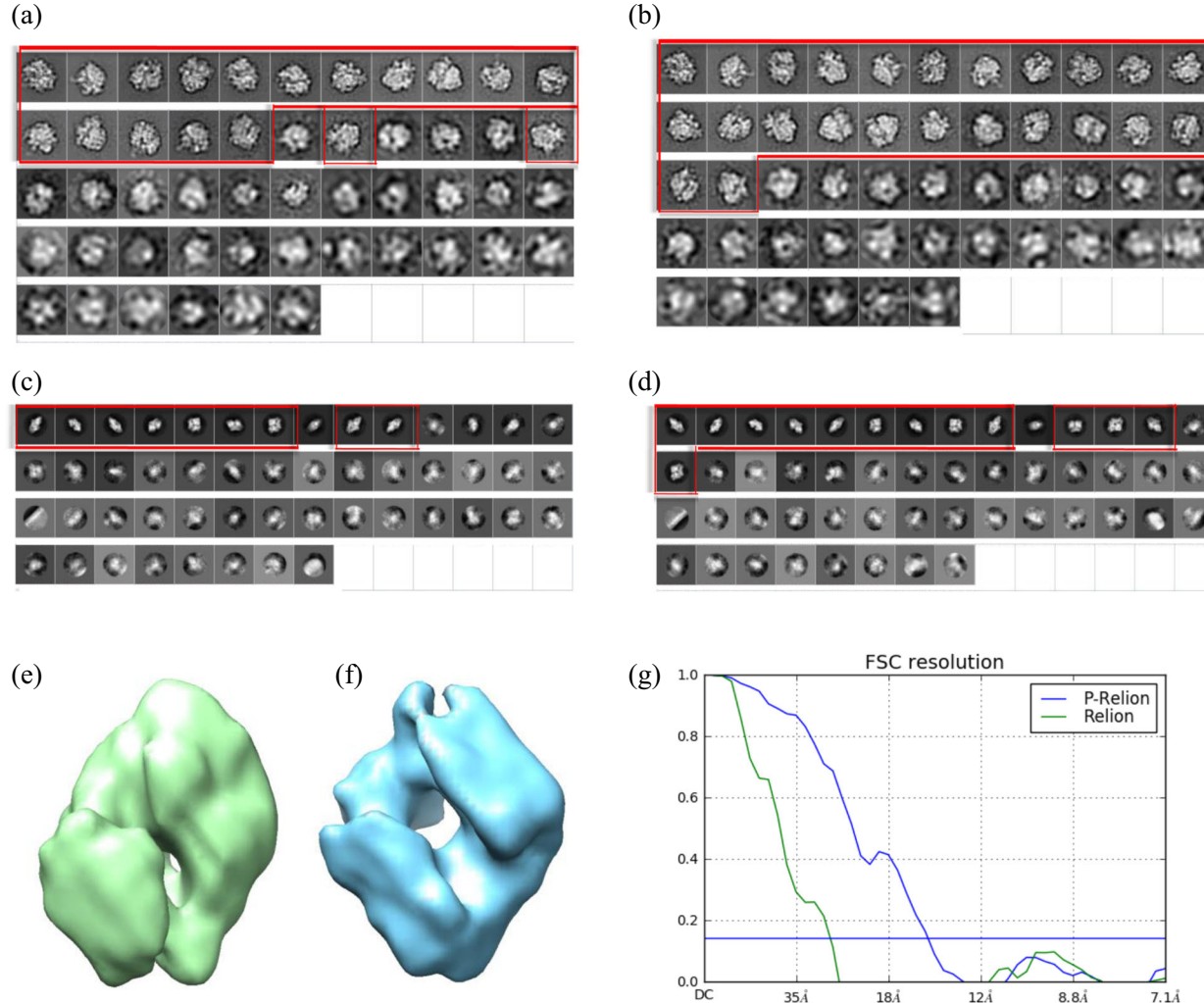

**Fig. 2 The RELION Classification Results of Two Small Datasets: 70S ribosome and beta-galactosidase. (a)** Classification 70S ribosome experimental cryo-EM images using RELION; 18 good classes are boxed in red. **(b)** Classification 70S ribosome experimental cryo-EM images using RELION with the aid of the pre-processing (P-RELION); 24 good classes are boxed in red. **(c)** Classification beta-galactosidase experimental cryo-EM images using RELION; nine good classes are boxed in red. **(d)** Classification beta-galactosidase experimental cryo-EM images using P-RELION. 13 good classes are boxed in red. **(e)** and **(f)** are the initial models calculated from the good classes in **(c)** and **(d)**, respectively, where the calculation was performed using PRIME; the one from RELION is in green and that from P-RELION is in blue. **(g)** shows the comparison of an initial model to the golden model using Fourier Shell Correlation (FSC) curve. The FSC curve of comparing (e) to the golden model is colored in green while that of comparing (f) to the golden model is colored in blue. The fitting of an atomic model to these two initial models is displayed in Supplementary Fig. 6.

datasets while leaving other parameters unchanged by the default values. When the pre-processing is used prior to ISAC, we abbreviate the procedure as P-ISAC. With the pre-processing, the number of stable classes is increased from 40 to 45 for 70S ribosome and from 37 to 41 for beta-galactosidase (Fig. 3). The pre-processing has increased the occupancy in many classes for both datasets, evidenced by the histogram in Supplementary Figs. 8 and 9a, d. The yield of particles for the 70S ribosome is increased from 78% to 88% while that for the beta-galactosidase is increased from 58% to 67%. Since ISAC was reported to have tendency to lose lower-defocused particles[11], we investigated the distribution of the defocus values of the harvested particles. Since each particle was labeled with a defocus value in this set, we searched the medium value to find that 2.5 μm was a good approximation. As shown in Supplementary Fig. 10, the yield of smaller defocus particles (<0.5 μm) from ISAC is 73%, lower than 85%—that of larger defocus particles (>2.5 μm). Interestingly, with the addition of the pre-processor, the yield of smaller defocus particles is increased to 81% while that of larger defocus

to 90%, suggesting the potential of the pre-processor in saving more closer-to-focus or lower contrast particles.

ISAC is known to offer high-quality classes[11] that one would not anticipate significant improvement of the initial model by the extra pre-processing. Nonetheless, we proceeded to calculate the initial models of the beta-galactosidase from the class averages. The findings show that the initial model from P-ISAC is better than that from ISAC (Fig. 3(e)).

Concerned about the time consumed by ISAC, we measured the duration spent on classifying those two small datasets to find that the pre-processor could help save the time on 2D classification by approximately 30–40% (Table 1), which results in a time-saving by 20% for the entire workflow because 2D classification consumes approximately 60% of the time of the whole workflow for the small datasets.

Using the beta-galactosidase particles harvested through 2D classification, we further performed 3D refinement using the initial model to find better final 3D results could be obtained with the pre-processor Table 1.

**Table 1 The classification time and the resolutions of the final 3D map on six datasets without versus with pre-processing.**

| Algorithm | Dataset | Original/with pre-processing | | |
|---|---|---|---|---|
| | | Execution time (h)[a] | Number of good classes | Resolution (A)@0.143 |
| RELION | 70S ribosome | 0.35/0.32 | 18/24 | N/A |
| | Beta-galactosidase | 0.27/0.28 | 9/13 | 13.31/8.89 |
| | 80S ribosome (520 classes) | 32.65/29.85 | 148/184 | 3.16/3.15 |
| | 80S ribosome (200 classes) | 18.81/18.08 | 91/99 | 3.13/3.12 |
| | 80S ribosome (100 classes) | 12.38/12.27 | 58/64 | 3.13/3.12 |
| | TRPV1 (175 classes) | 5.42/5.28 | 12/17 | 3.31/3.25 |
| | TRPV1 (100 classes) | 3.58/3.52 | 17/20 | 3.37/3.28 |
| | TRPV1 (50 classes) | 2.46/2.54 | 14/14 | 3.35/3.26 |
| | NC-TRPV1 | 13.56/13.52 | 15/20 | 3.57/3.42 |
| | NanoD-TRPV1 | 9.49/9.55 | 20/23 | 3.01/2.86 |
| | NanoD-TRPV1 (second pass) | */6.23 | */13 | */2.82 |
| ISAC | 70S ribosome | 2.85/1.89 | 40/45 | N/A |
| | beta-galactosidase | 1.58/1.14 | 37/41 | 7.73/7.47 |
| | 80S ribosome (4X down-sampling) | 124.50/105.35 | 514/520 | 3.10/3.10 |
| | TRPV1 (no down-sampling) | */39.65 | 0/67 | */3.80 |
| | TRPV1 (2X down-sampling) | 78.33/43.31 | 124/124 | 3.53/3.31 |
| | TRPV1 (3X down-sampling) | 47.96/37.33 | 156/156 | 3.30/3.20 |
| | NC-TRPV1 (3X down-sampling) | 10.56/8.48 | 25/27 | 3.56/3.39 |

Notice that 3D reconstruction is not conducted on 70S ribosome dataset and ISAC would fail without down-sampling or pre-processing in the TRPV1 dataset

[a]For RELION, the number of GPU used in 70S ribosome, beta-galactosidase and 80S ribosome is 3 while 1 is used for TRPV1 and NanoD-TRPV1. On the other hand, the number of Message Passing Interface (MPI) used in ISAC for 70S ribosome and beta-galactosidase is 10 while 44 is used for all the other datasets. Finally, the execution time is the average over 5 rounds for 70S ribosome and beta-galactosidase and 3 rounds for all the other datasets

**Pre-Pro on 80S ribosome is cost-effective and lossless.** Since the small datasets limit the resolution by the particle number and pixel resolution, or the quality of data, we set out to diagnose the pre-processor using large datasets that contain information of near-atomic resolution. To this end, we first chose two datasets: 80S ribosome (EMPIAR-10028)[16] and TRPV1 ion channel (EMPIAR-10005)[17], both contain a large number of good particles that were reconstructed to better than 3.5 Å to support the building of atomic models from De Novo.

The 80S ribosome particles, isolated from a malaria parasite (Plasmodium falciparum) and drugged with of emetine[16], are large and mostly rigid. This dataset contains a total of 105,247 particles and has been processed by RELION with the radiation damage issue compensated by B-factor weighting to report a structure with an average resolution of 3.2 Å (0.83 Nyquist), where the resolution of 40S subunit is lower than the average[16]. Due to the fact that the overall resolution is near the Nyquist limit, we do not expect significant advance on the attainable resolution from this dataset. Concerned about the computation cost of an algorithm on such a large set, we first measured the time spent on the pre-processing. The measurements reported that the denoising step and the 2D reference-free alignment step (Fig. 1(b)) only took 5 min and 50 min, respectively. In this section, we re-curated this large set with 2D classification and then calculated a 3D structure from the resultant particles using CryoSparc 3D refinement[18] guided by the initial model generated from the 2D class-average images.

To perform RELION classification on this 80S dataset, we let the prescribed number of classes to vary from 100, 200 to 520. It took a total of 12–32 h to complete RELION classification (Table 1). When the pre-processing was included, the number of clear classes from RELION was increased by roughly 10–20% (Table 1, Supplementary Fig. 11 and 12). It is noted that the increase of good classes does not remarkably increase in the yield of particles—in the case with the prescribed number of 100, the yield changes from 99.0 to 99.4%. We further performed 3D reconstruction using the harvested particles—in the original form neither down-sampled nor re-positioned for both cases of with and without the pre-processing.

As shown in Fig. 4(e) and Supplementary Fig. 13, the particle set obtained from P-RELION classification with 100 classes or 200 classes provides a 3D structure with an overall resolution of 3.12 Å (0.86 Nyquist).

To facilitate the test ISAC on the 80S ribosome dataset, we increased the limit of the class size to 200 and binned the images by a factor of 4 (4X down-sampling) to reduce the image to 90 × 90 pixels since the size of original 80S ribosome images is enormous. ISAC succeeded in classification such a large dataset and produced 520 stable classes, but this process took a total of 124 h. Interestingly, with the intervening of the pre-processing 20 h were saved (16% of the total time) (Table 1). In addition, the pre-processing helps ISAC produce only a few more stable classes from this large set (Table 1, Supplementary Fig. 14), while the change in the yield of particles is also insignificant—from 97.7% to 98.2% (Fig. 4(e): column 4). Regarding the final 3D structure, both ISAC and P-ISAC have led to the same overall resolution of 3.10 Å (0.87 Nyquist).

Since the quality of a cryo-EM map can vary from site to site, we use the local resolution method[19] to further evaluate the maps. As shown by the heat map provided by CryoSparc's local resolution program (it re-implements the blocres[20] program with GPU acceleration) (Fig. 4(h)), those best maps exhibit a broad range of resolutions, and report that most parts in each map are resolved close to or better than 3.0 Å (deep blue), whereas the low- (red) and medium-resolution (white) regions are sparsely distributed—most of them are localized to 40S subunit, highlighted by the lower-left corner in Fig. 4(h). Guided by the heat maps, we found noticeable modifications in density map could be introduced by the pre-processor to some flexible elements—for example, the protruded stalk of the 60S subunit and some parts in the 40S subunit.

In summary, the particle harvest tests on the 80S ribosome dataset demonstrate our pre-processing can help preserve virtually the entire set of particles of high homogeneity. Importantly, the computation cost of the pre-processing on this large dataset is extremely low—less than 1 h is consumed by the pre-processing, contrasted to tens of hours used by RELION 2D classification and much more by ISAC.

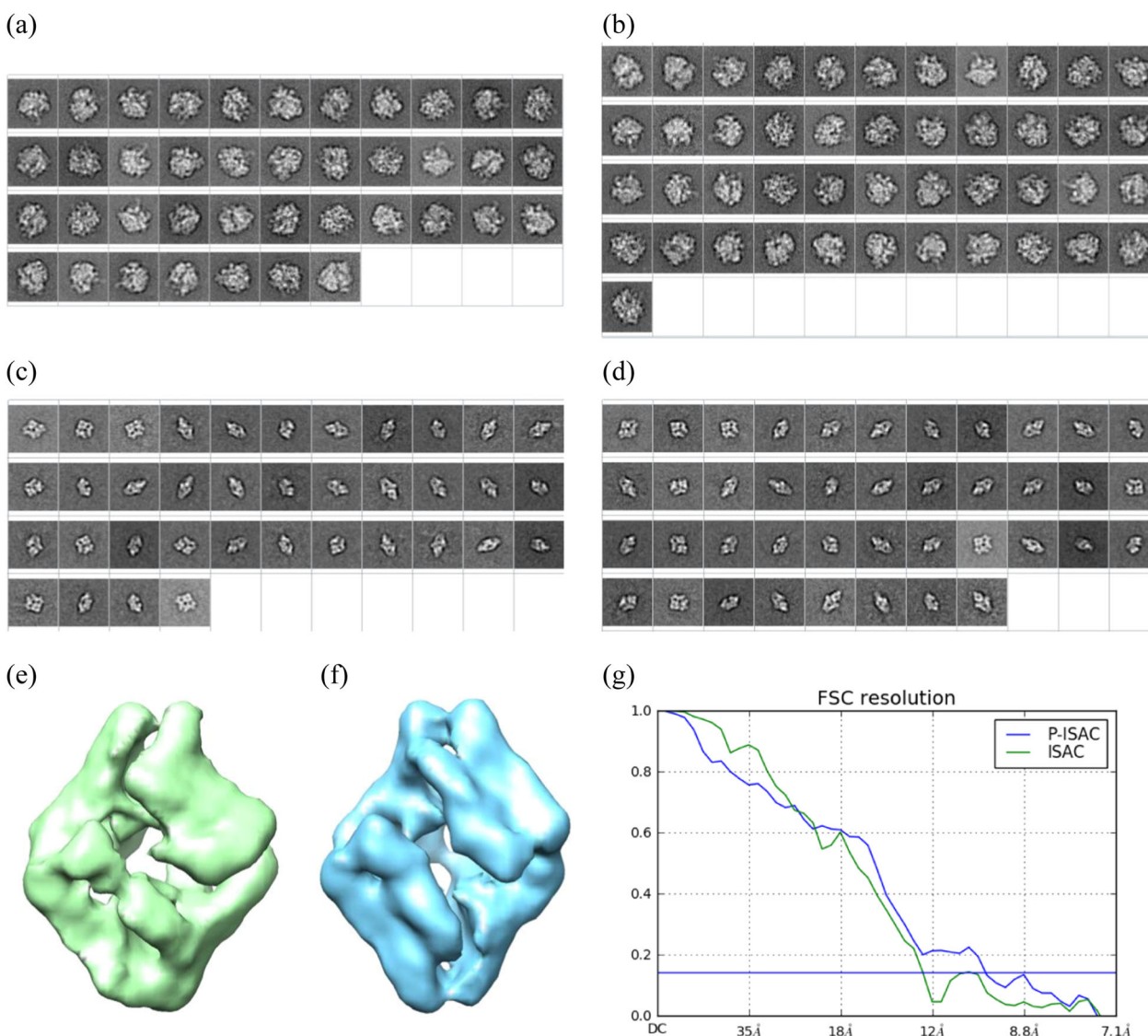

**Fig. 3 The ISAC classification results of two small datasets: 70S ribosome and beta-galactosidase.** (**a**) 40 classes of 70S ribosome are produced by ISAC. (**b**) 45 classes are produced by P-ISAC. (**c**) 37 classes of beta-galactosidase are produced by ISAC. (**d**) 41 classes of beta-galactosidase are produced by P-ISAC. (**e**) and (**f**) show the initial models calculated from (**c**) and (**d**), respectively, where the calculation was performed with PRIME. The initial model from ISAC is in green and the one from P-ISAC is in blue. (**g**) shows the comparison of the initial models to the golden model. The green FSC curve shows (**e**) to the golden model while the blue FSC curve shows (**f**) to the golden model.

**Pre-Pro enhances map interpretability of curated TRPV1.** The particles of TRPV1 ion channel, cloned from rat (Rattus norvegicus) and expressed by and purified from a human cell line[17], represent a tough dataset because the particle is smaller (300 kDa) where the protein feature is obscured by the amphipol molecules on the surface. The collection of 35,645 particles represents a highly curated set using 2D and 3D classifications and has reported a structure of 3.4 Å (0.70 Nyquist)[17]. It is noted that the resolution of this very set was extended to 3.3 Å by reprocessing using CryoSparc[18]. In this test, we re-curated this set with 2D classification and used the resultant particles to calculate a 3D structure using CryoSparc[18] guided by the class averages generated initial model.

To optimize RELION classification of the TRPV1 dataset, we screened three prescribed class numbers, 50, 100, 175. Compared to 80S ribosome (Fig. 4(e)), the yields of the TRPV1 particles vary considerably with the class number to exhibit a broad range of distribution (62–77% of 35,645) (Fig. 5(e): column 1–3), while the overall resolutions of the resulting 3D structures vary little with

the class number (Fig. 5(f): column 1–3). It is noted that the highest resolution—3.31 Å, virtually the same as that from full CryoSparc processing, was obtained from the least number of particles—approximately 22,000 (62% of 35,645) (Fig. 5(e): column 1), suggesting the existence of a finer subset. When the pre-processing was added prior to RELION, improvements in the overall resolutions are in the range 0.06—0.09 Å (Fig. 5(f): column 1–3; Table 1), lifting the resolutions beyond 3.3 Å. Interestingly, larger improvements were associated with the cases of lower resolutions (Fig. 5(f): column 2; Table 1).

We then tested ISAC classification on the TRPV1 dataset, by which we set the class size to 200. The run with ISAC had failed to converge that no class could be obtained (Fig. 5(e): column 4), yet this faltering could be rescued with our pre-processing, by which 67 classes (Supplementary Fig. 15) containing a total of 6,254 particles were produced (Fig. 5(e): column 4) to yield a 3D structure of 3.8 Å (Fig. 5(f): column 4). These classes are only half-filled and among them the top views are infrequently reported. To test if increased contrast would help restore ISAC,

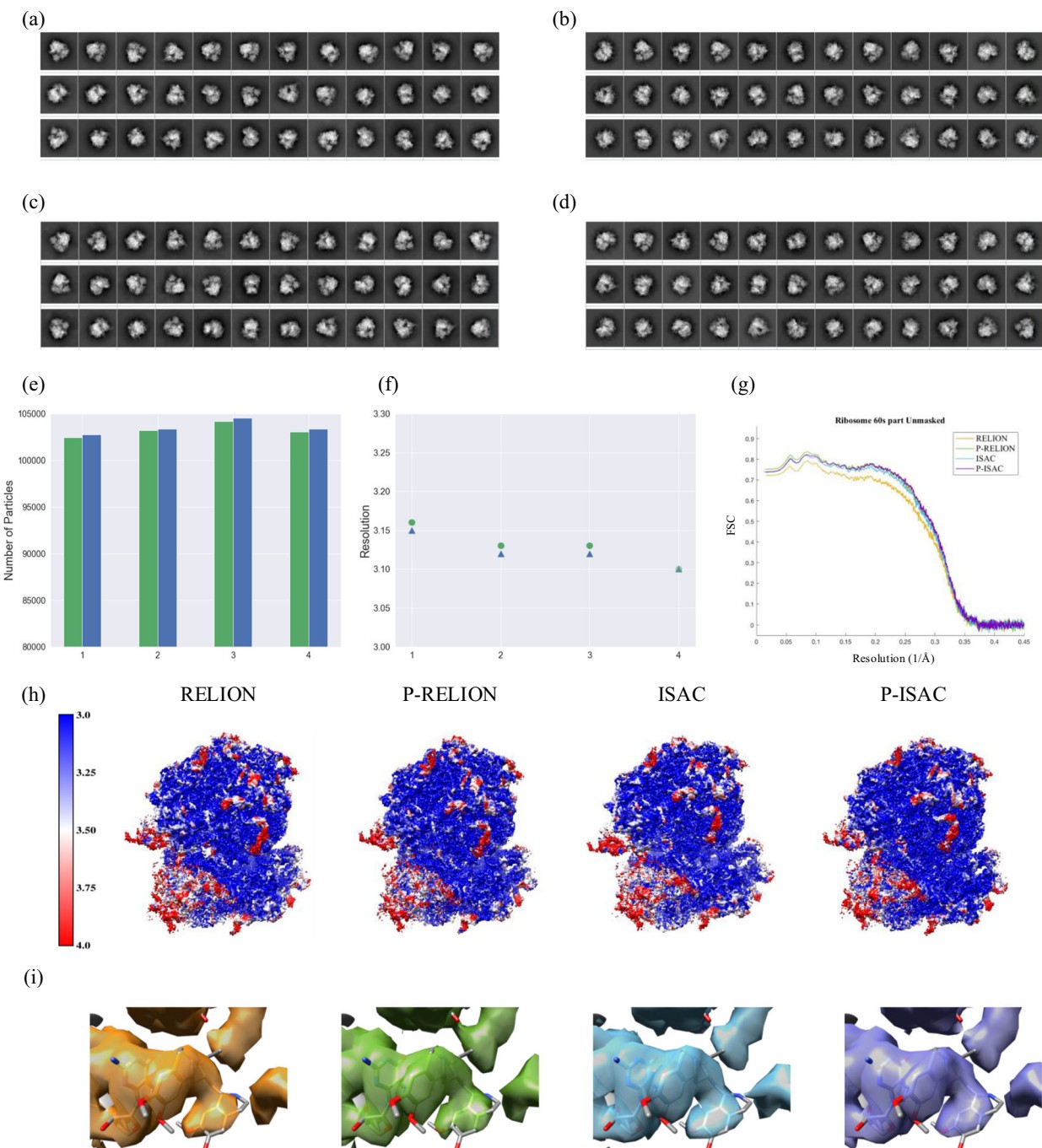

**Fig. 4 The Classification Results of Malaria (Plasmodium falciparum) 80S ribosome (EMPIAR-10028).** (**a**) Representative classes from RELION with 100 prescribed classes, and (**b**) the same as (**a**) except from P-RELION. (**c**) Representative class averages from ISAC, (**d**) the same as (**c**) except from P-ISAC (See Supplementary Figs. 12 and 14 for all the class averages). (**e**), (**f**) Summarize the statistics of the particle yields, the overall final resolutions. Four experiments under different settings are conducted (colored with green/blue). 1: RELION/P-RELION with 520 classes, 2: RELION/P-RELION with 200 classes, 3: RELION/P-RELION with 100 classes, 4: ISAC/P-ISAC with 4X down-sampling. (**g**) Shows the corresponding map-to-model (60S, PDB 3j79) FSC curves calculated using Phenix[50]. (**h**) Shows the overall density maps (threshold = 1.30) corresponding to 3 and 4 in (**f**) that are displayed according to the local resolutions using that re-implemented in CryoSparc. From left-hand side to right-hand side are the sharpen maps for RELION, P-RELION, ISAC, and P-ISAC with the B-factors of 126.4, 125.6, 124.7, and 124.4, respectively. (**i**) The densities of emetine.

we down-sampled the TRPV1 particle images by 2X. With this down-sampling, ISAC was rescued, producing more than a hundred of classes (Supplementary Fig. 15) that contained approximately 12,000 particles (Fig. 5(e): column 5). However, the success of this restoration cannot be attributed entirely to the increased contrast because the contribution from the confounding factor of reduced image dimension or others cannot be ruled out. When the 2X down-sampling was further aided with the pre-processing, the total number of classes remained the same, but the occupancy in each class was increased, yielding more than 20,000 particles (Fig. 5(e): column 5), approximately twice as many as those without the pre-processing (Supplementary Fig. 15). In addition, with the aid of the pre-processing (Table 1), the time of ISAC spent on the 2X down-sampled dataset was cut from 78 to

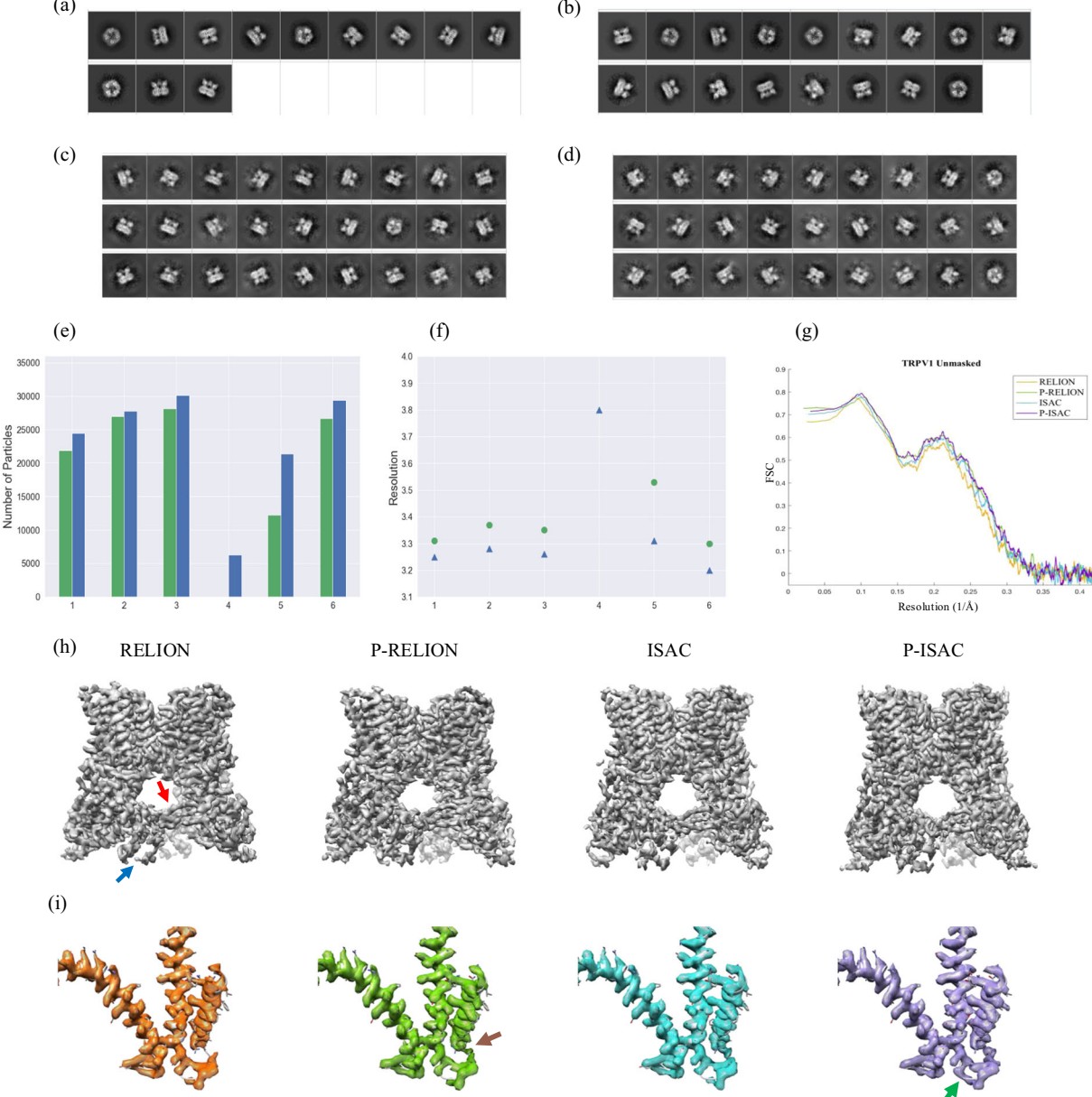

**Fig. 5 The classification results of Rat TRPV1 channel (EMPIAR-10005).** (**a**) Representative classes from RELION with 175 prescribed classes, and (**b**) the same as (**a**) except from P-RELION. (**c**) Representative classes from ISAC with 3X down-sampling, (**d**) the same as (**c**) except from P-ISAC (See Supplementary Figs. 15 and 16 for all the class averages). (**e**) The statistics of the total number of harvested particles. Six clustering experiments under different settings are conducted (colored with green/blue). 1: RELION/P-RELION with 175 classes, 2: RELION/P-RELION with 100 classes, 3: RELION/P-RELION with 50 classes, 4: ISAC without down-sampling, 5: ISAC/P-ISAC with 2X down-sampling, 6: ISAC/P-ISAC with 3X down-sampling. (**f**) The overall final resolutions corresponding to those in (**e**). (**g**) The map-to-model FSC curves by comparing the maps of RELION of 1, P-RELION of 1, ISAC of 6, and P-ISAC of 6 in (**e**) to PDB 3j5p. The calculation was performed using Phenix[50]. (**h**) The overall density maps of RELION of 1, P-RELION of 1, ISAC of 6 and P-ISAC of 6 in (**e**). The beta-sheets and the ankyrin repeats are indicated with a red arrow and a blue arrow, respectively. The B-factors measured and used for sharpening the four maps are 86.1, 90.1, 90.0, and 92.7, respectively. (**i**) enlarged densities corresponding to (**h**) where the same threshold of 0.49 on Chimera[51] was applied. The left helix corresponds to 559–628 of subunit C, displayed as a reference. The middle and the right helices correspond to 429–455 and 469–498 of subunit A and they form a "U" with the connecting loop (456–468). The brown arrow denotes 464–468 while the green arrow denotes 456–458.

43 h—a reduction by 45%, which results in 40% time-saving on the entire workflow. Finally, when P-ISAC was applied to the 3X down-sampled data, a 3D structure of unprecedented resolution was produced for this dataset—3.20 Å (0.75 Nyquist)(Fig. 5(f): column 6, Supplementary Fig. 15). Similar to RELION, the tests on ISAC show that larger improvements by the pre-processing are associated with the cases of lower resolutions (Fig. 5(f): column 2; Table 1).

By comparing these best maps, we found slight modifications in the density map of TRPV1 could be introduced by the pre-processor—for example, in the cytoplasmic region that includes the beta-sheets (red arrow in Fig. 5(h)) and the ankyrin repeats (blue arrow in Fig. 5(h)). Noticeable changes become evident when we zoom in the map—in a protein loop of 13 residues, from residues 456 to 468 (456–458), which exhibits gaps in the original map from RELION, the density of 464–468 is restored in the

P-RELION map (indicated by a brown arrow in Fig. 5(i)) and in the ISAC map as well (Fig. 5(i)), while that of 456–458 is further restored in the P-ISAC (green arrow in Fig. 5(i)).

Compared to the 80S ribosome, classifying this TRPV1 dataset of increased heterogeneity and lower contrast shows fine-tuning optimization parameters would lead to measurable improvements. Notably, when the pre-processor was added, more pronounced improvement was imparted on the less optimized cases to yield similar final results.

**Pre-Pro improves resolution of non-curated TRPV1 in nanodisc.** So far, all tests on large dataset have been restricted to curated datasets where marginal improvements were gained for the overall resolutions. We suspect that larger impact could be made by the pre-processor on non-curated datasets that have contaminants. To this end, we further tested two datasets, one is the non-curated full dataset of TRPV1 (EMPIAR-10005) and the other is a ligand-bound TRPV1 channel embedded in nanodisc (EMPIAR-10059). The non-curated set of TRPV1 is referred as "NC-TRPV1" and the nanodisc-embedded TRPV1 set as "NanoD-TRPV1". Compared to the NC-TRPV1 data where the later frames were eliminated[17], the radiation damage in NanoD-TRPV1 data was compensated by dose-weighting[21].

The original set of NC-TRPV1 downloaded from EMDB database contains slightly more than 80,000 particles. To perform 2D classification, we used 200 classes as the prescribed number for RELION and 3X down-sampling for ISAC as they were the best settings found for the tests on the curated set of 35,645 particles. RELION and ISAC give 50,620 and 43,690 particles respectively, yielding two structures with indistinguishable resolutions—3.57 and 3.56 Å (Table 1 and Supplementary Figs. 15 and 16). The pre-processor increases the respective number to 55,269 and 52,661 and furthers the resolutions to 3.42 and 3.39 Å respectively. These resolutions are comparable to that reached by curating the original set with 2D classification followed by 3D classification[17]. We noticed that another pass of P-RELION gave 42,868 particles and extended the resolution to 3.37 Å.

The NanoD-TRPV1 downloaded from EMDB database contains 218,787 particles, from which the authors selected 73,929 particles using 2D and 3D classifications to obtain a final structure of 2.95 Å (0.82 Nyquist)[21]. For this dataset, we used RELION (3.0)[15] to speed up the tests since the size of this set is enormous—it is the largest set in this study. We set the prescribed class number to be 100 to save time on 2D classification. With this setting, RELION (3.0) 2D classification was finished within 10 h. Typical with RELION, without and with the pre-processor, RELION sifted similar fractions of particles—70% (153,839) and 76% (166,236) respectively. However, the resolutions of the resulting structures differ substantially—3.01 versus 2.86 Å with the latter obtained through P-RELION (Supplementary Fig. 17). When additional run of P-RELION was applied to the set of 166,236 particles, the resolution was extended to 2.82 Å (0.87 Nyquist). As we compare the RELION map with two P-RELION maps (Fig. 6), improvement of the maps is evident in the cytoplasmic part including the ankyrin repeats. In summary, the tests on non-curated dataset demonstrate the potential of the pre-processor in making larger impact on more heterogeneous data.

## Discussion
2D classification plays a key role in processing single-particle cryo-EM images—it is mainly used to curate a particle set, but is faced with challenges due largely to the noisy nature of the data. In this report, we introduce a fast and effective pre-processor that has a built-in particle denoiser to enhance the performance of 2D classification. One novelty of this tool lies in its two-step

approach. In the first step, the particle shape information is boosted by denoising through 2SDR a method that is much faster than those existing PCA approaches[22,23]. In the second step, the original particles are re-positioned using the parameters extracted for aligning the denoised "high-contrast" particles. This approach prepares a better aligned input set for 2D classification and consciously avoids the consequence from possible information corruption by the denoising. By implementing the pre-processor into software package RELION, ISAC and CL2D[10] (Supplementary Figs. 18–22) with tests performed on several benchmark sets, we demonstrate this processor with minimally added computation cost can make 2D classification faster, improve the yield of good particles, and increase the number of good classes to give rise to better initial models, particularly evidenced by the beta-galatcosidase particles. In addition, we found the pre-processor can save more closer-to-focus particles, suggesting its potential for benefiting lower contrast data. Further reprocessing the harvested particles of large datasets to generate 3D density maps is enabled by integration of the pre-processor into the entire processing workflow (Fig. 1). In the step of 3D refinement, we used CryoSparc homogeneous refinement for its rapidity, but initialized this step with the initial model produced from 2D class-average images. Surprisingly, reprocessing the large datasets with the pre-processor can further improve the final resolutions to surpass previous works: 3.10 Å for the malaria 80S ribosome, 3.20 Å for the TRPV1, and 2.82 Å for the TRPV1 in nanodisc. When the pre-processor is applied to RELION 2D classification on cleaning non-curated datasets, the resultant improvements are larger than those on curated datasets—the scale of improvements is as large as 0.2 Å. This scale is notable as it is comparable to that made by switching from MotionCorr to MotionCor2[24] or by compensating radiation damage using Unblur[25].

In the wake of cryo-EM resolution revolution, furthering the resolution by single-particle approach to that offered by X-ray crystallography would benefit pharmaceutical applications. Recently, Bartesaghi et al.[26] demonstrated a near-atomic cryo-EM structure of beta-galactosidase could be extended to atomic resolution (1.8 Å) by adding three more correction steps to the processing workflow, yielding an overall advancement of 0.4 Å, which is a combined effect of three measurable increments: 0.07, 0.09 and 0.12 Å obtained by correcting local defocus, local drift and radiation damage effect respectively. In this view, the improvement of 0.2 Å made by the pre-processing is significant. It should be noted that, given a B-factor of 100 Å$^2$, if one wishes to improve the resolution by 0.2 Å in sub-3 Å regime by taking more data, one would need to increase the size of the dataset by at least three times[27].

It is seemingly bewildering that better classification results can be obtained with the pre-processing since it merely re-positions the original particles without increasing their SNRs. To possibly explain this puzzle, we consider 2D classification as an optimization problem—it searches for an optimal solution in a very high dimensional space whose parameters include the alignment parameters of all the particle images. Usually, a search algorithm iterates until convergence occurs and is likely to be trapped at a local minimum, in particular when it faces high dimension or non-convex problems. It is well known when the solution of global optimum is not guaranteed, initial values can drastically impact the final solution[28,29] for which strategies of setting best initial values are being actively developed. In this light, our finding implies that the pre-processing may have set better initial values to predestine the investigated classification algorithms to escape from some, if not all, local minimums[30]. Given so, it explains why the pre-processor can benefit RELION, ISAC, CL2D[10] (Supplementary Table 1, 2 and Supplementary Figs. 18–22) and even CryoSparc (Supplementary Fig. 23), and

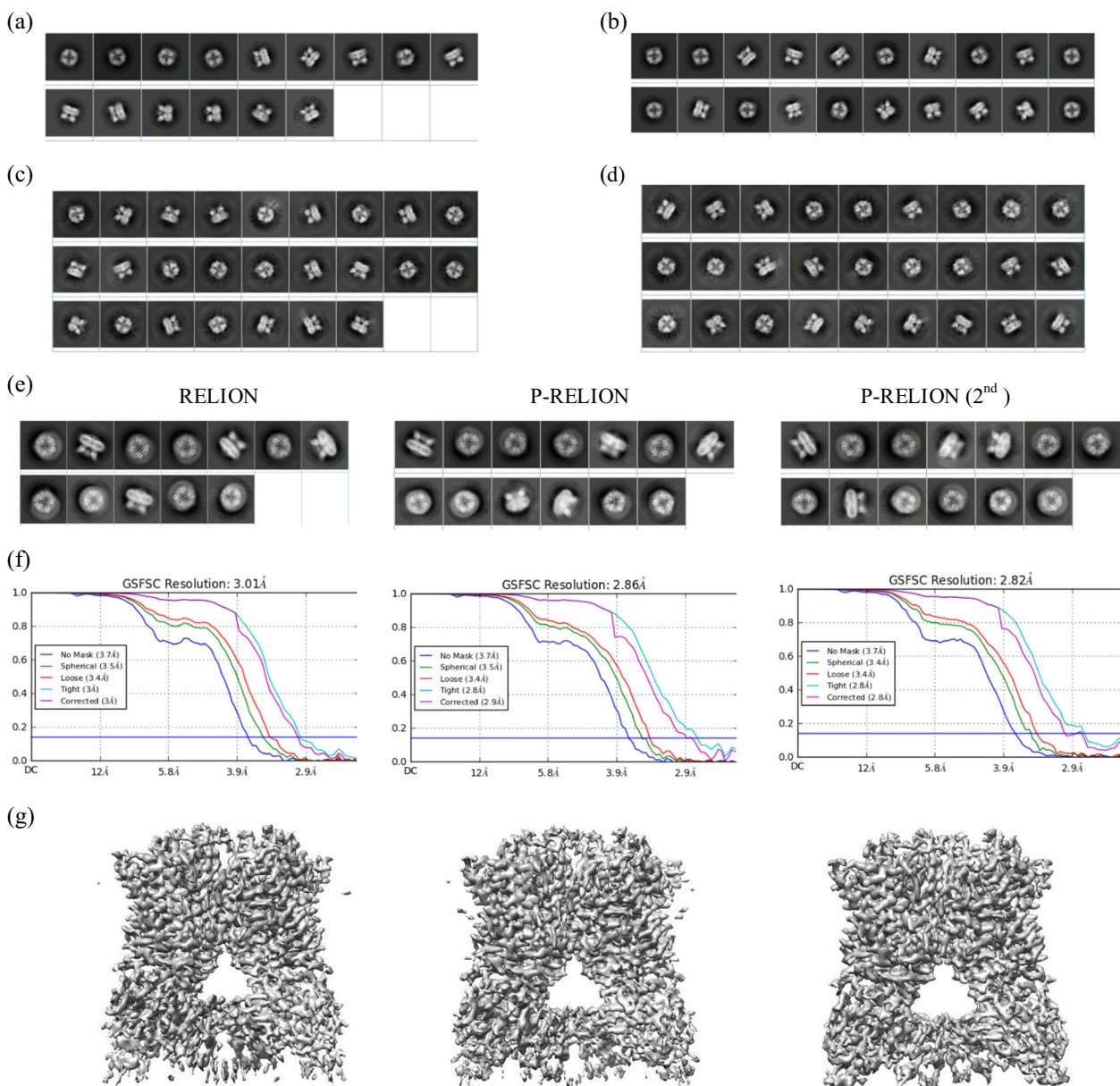

**Fig. 6 The Classification Results of NC-TRPV1 (EMPIAR-10005)((a)–(d)) and NanoD-TRPV1 (EMPIAR-10059)((e)–(g)).** (a) Representative classes of NC-TRPV1 from RELION with 200 prescribed classes and (b) from P-RELION. (c) Representative classes from ISAC with 3X down-sampling on NC-TRPV1, (d) from P-ISAC (See Supplementary Figs. 15 and 16 for all the class averages). For (e)–(g), from left to right are the results of RELION, P-RELION, and P-RELION (second pass). (e) Representative classes of NanoD-TRPV1 using RELION or P-RELION with 100 prescribed classes (See Supplementary Fig. 17 for all the class averages), (f) the FSC resolution for NanoD-TRPV1, (g) shows the overall density maps (threshold = 0.55). The B-factors measured and used for sharpening the three maps are 130.6, 123.5, 117.3, respectively.

suggests the pre-processor may benefit other popular packages that also rely on K-means for clustering[22,31] or use expectation-maximization[32]. Since the pre-processor is modular in nature, its usage is not restrictive to the 2SDR denoising, but can be adaptable to any other denoising methods—for example, neural-network (NN) denoising methods which have been recently used with good results for particle picking[33–36].

## Methods
**Conceptual framework of the pre-processing**. We describe our scheme of denoising cryo-EM images and its rationale, but present the rigorous mathematical framework in a separate publication[12]. This scheme is based on a dimension-reduction method that has its roots in principal component analysis (PCA). This type of approach, first introduced to single-particle EM in the form of multivariate statistical analysis (MSA)[37], has been used to assist 2D analysis of much noisier cryo-EM data[38]. However, because PCA needs to solve the eigenvectors of co-variance matrix of vectorized images, the complexity of computation is scaled up with the square of the number of pixels of an image. As a result, the usage of PCA on large images is unfavorable[22]. To remove the bottleneck rooted in treating an image as a vector, alternative approaches were proposed to keep an image as a matrix[39–41]. Based on such notion, a two-stage dimension-reduction method (2SDR) was introduced to reduce the computation cost[12]. Additionally, since matrix vectorization would lead to extreme high-dimensionality, which together with low SNR, make the eigenvector estimation of PCA unreliable, we first treat the image as a matrix and employ matrix matrix rank reduction method like Higher Order Singular Value Decomposition (HOSVD)[39] or multilinear principal component analysis (MPCA)[40,41] to reduce the matrix into smaller dimension. Observing that the reconstructed cryo-EM particle images by HOSVD or MPCA still carry heavy noise, we introduce a PCA model in the second stage that vectorizes the reduced-rank matrix and clean another layer of noise. In this work, 2SDR (HOSVD-PCA) is performed on the images with $(p_0, q_0) = (25, 25)$

components in the HOSVD step and $r = 50$ in the PCA step (we use HOSVD instead of MPCA to speed up the computation and we use fixed rank instead of the rank selection procedure described in ref. [12]). To highlight the SNR improvement by the 2SDR denoising, we summarize the SNR of the original image and that of the denoised image for all the test data in Supplementary Table 3.

Since the initial references for alignment in a 2D classification algorithm are randomly selected from the entire set of raw images, which are very noisy, the resulting best matching class members could be unrelated or the best alignment parameters erroneous at the beginning. To mitigate the effect of the noise, we first employ 2SDR to the raw particle images. The denoised image dataset is then subjected to a fast reference-free alignment procedure, from which we record and extract the alignment parameters of rotation angles and $x$-and-$y$-direction shifts. To alleviate the consequence from the loss of high-resolution structure information through dimension-reduction, these alignment parameters are applied to the set of original images, not the denoised set. This procedure, as illustrated in Fig. 1(b), poses the images in new positions, ready to be fed to a 2D classification algorithm. As a result, the information of the original images remains intact except for minute effect from interpolation.

The reference-free alignment used in this work is based on that implemented in the SPARX package[42], which is much faster than the original algorithm[43]. The implementation of reference-free alignment is conducted in two stage. The first stage is to form a global approximation of image dataset by averaging all images into a single reference image. In the second stage, each image is compared with this reference image by varying the $x$–$y$ shift and the rotation—the parameters that give the highest cross-correlation are recorded. Finally, these parameters are apply to each image and the transformed images are averaged to form the new reference image. The process is repeated for several iterations to refined the alignment parameters. For all the experiments in this study, three iterations of reference-free alignment are applied on the denoised images and two additional iterations of reference-free alignment are used to fine-tune the re-positioned original images, as illustrated in Fig. 1(b). To facilitate the usage of the pre-processor, we provide a plugin that permits its insertion between particle picking and 2D classification (Fig. 1(b)).

**Data description and preparation**. In this study, six experimental cryo-EM datasets are used for the tests. The first set is a subset of 10,000 E. coli 70S ribosome, a widely used test dataset[6,10,14,44–47] recorded in Joachim Frank lab using a CCD camera (available from ftp://ftp.ebi.ac.uk/pub/databases/emtest/SPI-DER_FRANK_data). The original 70S ribosome set containing 91,114 boxed-out particles with box size of $130 \times 130$ pixels of 2.82 Å as previously described[46] can no longer be found from Electron Microscopy Data Bank. We only used the first half set, which is ribosome bound with an elongation factor (EF-G). We term it "70S ribosome". Each particle has the defocus parameters annotated (in the range of 2.5–3.5 μm). For the pilot and the classification experiments, we did not use the CTF-corrected particles, but used them when performing 3D reconstruction with CryoSprac[18] for a simulation study.

The second set contains 15 movies of beta-galactosidase recorded on a direct electron camera (Falcon II). It was downloaded from RELION 2.1 tutorial data. To deblur the micrographs, we performed motion correction using the movie frame alignment function on XMIPP[48]. The contrast transfer function (CTF) estimation was performed by Ctffind4[49] and the correction was applied to each micrograph. Finally, using XMIPP auto-picking function, we extracted 5672 particles from these micrographs with a box size of $100 \times 100$ pixels of 3.54 Å due to 2X down-sampled in RELION Tutorial.

The third dataset is a malaria 80S ribosome[16] from Scheres lab. It is downloaded from EMDB (accession number EMPIAR-10028). This set contains 105,247 particles carefully selected from a larger set of 158,212 particles by the authors[16] and is provided as a RELION Benchmark example. This data was recorded on a direct camera (Falcon II) using a 300 kV Titan Krios cryo-EM at MRC and it is used for testing the performance of a cryo-EM image processing algorithm. The size of the particle image is $360 \times 360$ pixels of 1.34 Å. With the defocus parameters provided for each particle by the authors, we applied CTF correction.

The fourth (curated) and fifth (non-curated) dataset are TRPV1 channel downloaded from EMDB (EMPIAR-10005). The original dataset was recorded on a direct electron counting camera (K2) with super-resolution mode (0.6 Å) in Yifan Cheng lab at UCSF using a 300 kV cryo-EM with a side-entry holder[17] and then extracted and decimated by a factor of 2 into images with a box size of $256 \times 256$ pixels (1.2 Å) by the authors. The 35,645 particles were carefully selected using 2D and 3D classifications by the authors from a larger set of 88,915 particles. We performed CTF correction using the provided defocus parameters.

The last one is TRPV1 embedded in nanodisc downloaded from EMDB (EMPIAR-10059). The dataset is with a box size of $192 \times 192$ pixels (1.2 Å) by the authors[21]. All the 218,805 particles were used in this study. We performed CTF correction using the provided defocus parameters.

**3D reconstruction**. As described in the workflow in Fig. 1(a), once a 2D classification process is completed, we use good class averages to calculate the initial model using the ab initio method provided by PRIME[14]. The initial model is then used to guide 3D refinement by CryoSprac. Note that to alleviate the loss of high-resolution information due to interpolation, we use the original set of

particles for 3D refinement, instead of the re-positioned set used for 2D classification. To assess the overall resolution of a structure, we use the standard Fourier shell correlation (FSC). For displaying local resolutions, we use CryoS-parc's local resolution program.

**Computation resources**. As for the computers used for the computation, all the experiments are run on a workstation equipped with two Intel Xeon CPU E5-2699 v4 at 2.20 GHz and eight NVIDIA Geforce GTX 1080 Ti graphics cards, except early pilot tests were performed on a notebook.

**Reporting summary**. Further information on research design is available in the Nature Research Reporting Summary linked to this article.

## Data availability

The 70S ribosome can be found in EMDB test image data. The beta-galactosidase can be downloaded from RELION 2.1 tutorial data. 80S ribosome is from EMDB (accession number EMPIAR-10028). TRPV1 channel is available in EMDB (EMPIAR-10005). TRPV1 embedded in nanodisc is available in EMDB (EMPIAR-10059). The 3D maps in Figs. 1–6 are available in https://drive.google.com/file/d/1iihyVw1Jy7ob9fKcq3ewkfZ8350qPFRM/. Remaining data are available from corresponding author upon reasonable request.

## Code availability

To facilitate the usage of 2SDR pre-processing, we offer a plugin to allow it to be installed prior to a classification algorithm or inserted into a cryo-EM image processing pipeline. The plugin, including source code, is available for noncommercial use as a download at http://sabid.stat.sinica.edu.tw/.

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

## Acknowledgements
I.-P.T was supported by Ministry of Science and Technology [MOST 106-2118-M-001-001-MY2] and an Academia Sinica Grand Challenge Project grant [AS-GCS-108-08]. W.-H.C. is grateful for long-term support on the development of a Cryo-EM platform at Academia Sinica through AS Nano Program, and AS SUMMIT Project supported by [AS-SUMMIT-107], [AS-SUMMIT-108], [AS-SUMMIT-109] and [MOST-107-0210-01-19-01],[MOST-108-3114-Y-001-002], [MOST-109-0210-01-18-02].

## Author contributions
I.-P.T. and W.-H.C. conceived and designed the project; S.-C.C. and B.-Y.N. wrote the 2SDR codes and conducted the classification experiments; S.-C.C., H.-H.L., and W.-H.C. analyzed the classification results; S.-C.C., H.-H.L., S.-H. Huang, and W.-H.C. analyzed the density maps; S.-C.C., I.-P.T., and W.-H.C. wrote the manuscript.

## Competing Interests
The authors declare no competing interests.
