## [Peer Review File · Communications Biology]

Reviewers' comments:

Reviewer #1 (Remarks to the Author):

This manuscript by Chung et al. described a procedure to enhance "2D clustering" of single particle cryo-EM particle images. The pre-process using a method "2SDR" that was described in a separate and published paper to reduce noise (denoise) of particle images, so that 2D classification of such denoised images produced better results. Overall, this is an interesting procedure, and could be potentially interesting and useful to cryo-EM practitioners. One specific result, reducing the time for ISAC 2D classification, would be interesting and useful to deal with heterogeneous dataset. ISAC is known to be very computationally expensive. Using the pre-process procedure described here, it would make the ISAC more applicable. Overall, this is an interesting manuscript.

Authors demonstrated the improvement of 2D classification by using pre-processing procedure (2SDR) that was described in a previous publication. Reading from the manuscript, authors emphasized the importance of pre-processing by 2SDR, which was already published. Intuitively, I would imagine that any improvement of SNR of particle images, by whatever method, would lead to a better classification. If so, intuitively, the procedure described here can be simplified as to state that improving SNR of particle images helps better 2D classification. For example, deep learning convolutional neural network denoise could potentially also lead to the similar improvement. I would suggest author to provide clearer description if this improvement of 2D classification is a result of improved SNR, or specifically by 2SDR? Regardless of the result, since 2SDR is already published, it would not reduce the potential interest of this manuscript.

It would be nice to have quantitative estimation of SNR improvement by the 2SDR pre-processing procedure. This could be done by quantitating spectra SNR of original raw image versus images after 2SDR denoising procedure.

The statement "confirmed by cryo-EM experts" is very subjective. Instead, the success of the procedure should be evaluated by quantitative measurement. In this case, one possible parameter is to compare the total number of particles within the "good" classes selected. If the number in (b) is more than that in (a), the procedure at least rescued more particles. While this is not entirely objective, it at least provides some "semi" quantitative measurement that is better than "confirmed by cryo-EM experts".

Minor:

"Clustering" is a rather vague and confusing term. My understand is that this is a 2D classification procedure. I suggest authors to change this to "2D classification", which is more accepted term.

Yifan Cheng

Reviewer #2 (Remarks to the Author):

In this manuscript Chung et al. proposed a pre-processing step to facilitate 2D clustering in cryo-EM data processing. It is a PCA based algorithm which can de-noise particle images fast. Implemented into software package Relion or ISAC, it provided potential improvements in 3 aspects. First, it could make data processing faster. Several benchmark data sets were tested with, and showed improvement in some cases. Secondly, it could improve the quality of class-average images, which could be checked visually, or reflected by a better initial model generated from these images. Lastly, it could increase the yield of good particles, which improved the final resolution by 0.03 - 0.09 Å; with Relion and 0.1-0.2 Å with ISAC.

The proposed algorithm is novel in cryo-EM data processing and can be implemented easily. However, the major benefit of the pre-processor is to increase the speed of 2D clustering using ISAC. It doesn't show convincing improvements when used with Relion, which has solved the largest number of reconstructions deposited in EMDB. It's unclear whether it will be helpful when using with other widely-used software, e.g., CryoSparc, which runs faster than Relion and ISAC, and produced reconstruction with the highest resolution to-date, or EMAN2, which has neural network features for better performance.

Queries:

1. There's little improvement of speed when using the pre-processor with Relion, as summarized in Table 1. The authors suggested that it was because Relion would run a fixed number of iterations based on the parameter given. Supplement Fig 4 shows that less iterations are needed with the pre-processor and thus save time. However, the claimed visual improvements between the two groups of figures are unclear.
2. The pre-processor has been tested with Relion2. However, Relion3 published in 2018 is 1.5-2.5x faster than Relion2. Can the pre-processor be coupled with this newer version and if so, how much improvement in speed will it provide?

3. Supplement Fig 5, which supports statistical improvement in Relion, is not clear. All the figures in column 1 and 3 shares the same X-axis . Each figure represents one experiment, and two data points were shown in each figure. The 2nd data points were X=999 in all figures, with explanation for the meaning of value 999. However, the X values for the 1st data point were not consistent, being 1, 3 or 4 in each figure. Similarly, figures in column 2 and 4 have X = 1, 2, 3 or 5. Why are these certain values chosen? When considering all data points for each experiment, does the statistics still show improvements?

4. The authors showed that their pre-processor reduced time spent on 2D clustering using ISAC by 30~40%. 2D clustering is one of the seven major steps for data processing. What percentage of improvement is achieved when considering the whole work flow?

5. Are the data used here damage-compensated or not? Damage compensation also de-noise particle images, which could have a similar improvement as the pre-processor. It would be helpful for the readers to know whether the pre-processor work with damage-compensated data.

6. Fig 2e, 2f, 3e and 3f show initial models were improved. The authors suggested that they were proof of higher quality of class averages thanks to the pre-processor. However, quality of any model is also based on numbers of particles or class averages, which were not consistent in these examples. Therefore, these models show that the pre-processor can help with the generation of initial model, which is beneficial for the whole work flow, but not necessary to be related to better class averages.

7. Improved local resolution shown in Fig 4h is hard to see as most parts of the figures are in blue color. Scaling the coloring differently or zooming in to the desired region could be helpful.

8. Reported resolution improvement is 0.03 - 0.2 Å; , which are not significant. This could be that the two small data sets are limited by number of particle, and the 80s ribosome data had reached the highest possible resolution determined by Nyquist frequency. Maybe different data should be tested to show the potential improvement from filtering.

Reviewer #3 (Remarks to the Author):

The manuscript by Chung et al. is well written and easy to follow. It proposes a modified

workflow for processing single particle cryo-EM data which enhances 2D clustering of particles and improves the resolution of reconstructed volumes for a variety of test datasets. This is achieved by adding a preprocessing step that uses a denoising algorithm (2SDR). All tested cases benefit from the preprocessing step with a minimal added computational cost. Also, the generation of the initial reference model is improved with the use of preprocessed data.

My main concerns are:

2D clustering is mainly used to clean Cryo-EM datasets and separate real particles from non-particles or contaminations. None of your 2D classes showed like contamination features. This seems normal given the nature of curated test datasets, but how the preprocessing step will perform in the presence of increased heterogeneity and contaminations?

Although all tested cases benefited from the preprocessing step, it seems from your results that the number of classes in Relion or down-sampling in ISAC have a bigger impact on the results than the use of the proposed additional preprocessing step. Given this, 2D clustering with optimized parameters such as the number of classes, regularization parameter T , angular sampling for Relion may not benefit as much from the preprocessing step. In some conditions, for the tested datasets, improvement is marginal.

Do you expect that datasets/particles closer to focus would be the ones that benefit more of the preprocessing step?

These and other comments are better explained in the attached review file and need to be addressed.

Thank you for the stimulating manuscript.

The manuscript by Chung et al. proposes the use of a preprocessor (2SDR; denoising algorithm) as an additional step for the common workflow for processing cryo-EM SPA data to enhance 2D clustering with a relatively minimal added computational cost. The use of the preprocessor increased the number of well defined clustered classes and with this the number of selected particles, producing maps of higher resolution and in some cases an improvement of the map interpretability and agreement of the maps with the models.

The manuscript is well written and easy to follow, it includes proper background of the basics and the

general workflow of cryo-EM data processing.

Comments

1. Number of classes in Relion or down-sampling in ISAC seem to have bigger impact on the results than

the use of the proposed additional preprocessing step (Table 1). Given the influence of the number of

classes in Relion, how would you expect changing parameters such as regularization parameter T and

angular sampling will play with the proposed preprocessing, are they still going to benefit in the same

proportion/amount.

2. Analyzed datasets comprise diverse test cases but is still remain an interesting question of how the preprocessing workflow will affect in cases of high heterogeneity or embedded particles in other structures

such as nanodiscs ?

3. The main common purpose of 2D clustering is to classify and clean particles from non-particles.

Heterogeneity and damaged particles are commonly treated with 3D classification, but not only. In

your case none of the classes show strong features of typical contaminations, current test datasets were

testing clustering mainly with good particles. This makes sense that given the nature of the datasets

they present little of no contaminations since they were curated and some are used for Relion tutorial.

Do you expect a similar improvement on classification with contaminations?

4. 2SDR denoising algorithm represents an essential part of the preprocessing proposed workflow, but

in the current state of the manuscript it is not clear what makes 2SDR denoising a better option to

denoise particles for clustering classification over other algorithms such as CWF[2] and neural network

denoising algorithms which have been recently used with great results for particle picking in packages

such as Warp[4], Topaz[1] and Cryolo/Janni[5, 3].

5. Starting references, generated with PRIME, use its own generated reference for each 3D refinement

or the same reference for all cases? what is the proportional contribution of starting with a

better reference? if any, or in other words does the quality of the initial reference affects results on top of the clustering? (particularly for poorly generated references). Was PRIME selected for any particular reason? In a recent article of Topaz-denoising[1], it is mentioned that for their case using denoised particles with Cryosparc 2 for Ab-initio model generation, produced less reliable results than using the raw particles. In your case, will results of Ab-initio model generation vary with different packages?

1

6. Do you expect that datasets/particles closer to focus would be the ones that benefit more of the preprocessing step? Can you comment on this with your current test cases.

7. It is mentioned in the manuscript that with the preprocessing step, convergence of 2D clustering is

faster. What criteria or parameters were used to monitor convergence of clustering? When you mention

you used up to 15 cycles for original data and 10 for preprocessed, had this already converged? If not,

will giving proper time for both to converge, would this close the gap between the results?

8. Figure 1a does not contribute for the purpose of explaining the modified workflow with the added

preprocess step, prepossessing step could be added as a brach in a different color, to help the reader

identify where it takes place and how its different from the classical workflow. This will also help to

link the workflow with figure 1b. Some steps are missing arrows for posible looping steps.

9. Figure 1b may give the sensation that a 3D refinement is performed on denoised particles and then

substituted by raw particles for another 3D refinement, but if I understood correctly the first referencefree alignment is a 2D and thus it needs explicit specification where a 2D or a 3D reference-free

alignment is performed. What is the contribution of extracting alignment parameters and transfer

them to the original images (are they really applied or only transfered to the apropiate files. Are the

2D alignment parameters used to reextract recentered images, if so, this should be better explained). If alignment parameters are not transferred to the original images, but only selection based on clustering is made, how much this affects currently reported results.

10. Figure 5 would benefit from a better labeling of graphs and volumes. In caption of image, for e) f) it is mentioned a map to model FSC but it seems that only g) shows FSC curves. It needs better explanation or rephrase. For i) what is the threshold level for displayed volumes? is the same for all of them?

11. All figures could benefit from a better indication/grouping in the figure to recognize quickly which classes and volumes come from original data, preprocessed data, Relion or ISAC.

References

- [1] Tristan Bepler, Alex J. Noble, and Bonnie Berger. Topaz-denoise: general deep denoising models for cryoem. *bioRxiv*, 2019.
- [2] Tejal Bhamre, Teng Zhang, and Amit Singer. Denoising and covariance estimation of single particle cryo-em images. *J Struct Biol*, 195(1):72–81, 07 2016.
- [3] Jaakko Lehtinen, Jacob Munkberg, Jon Hasselgren, Samuli Laine, Tero Karras, Miika Aittala, and Timo Aila. Noise2noise: Learning image restoration without clean data, 2018.
- [4] Dimitry Tegunov and Patrick Cramer. Real-time cryo-electron microscopy data preprocessing with warp. *Nat Methods*, 16(11):1146–1152, 11 2019.
- [5] Thorsten Wagner, Felipe Merino, Markus Stabrin, Toshio Moriya, Claudia Antoni, Amir Apelbaum, Philine Hagel, Oleg Sitsel, Tobias Raisch, Daniel Prumbaum, Dennis Quentin, Daniel Roderer, Sebastian Tacke, Birte Siebolds, Evelyn Schubert, Tanvir R. Shaikh, Pascal Lill, Christos Gatsogiannis, and Stefan Raunser. Sphire-cryolo: A fast and accurate fully automated particle picker for cryo-em. *bioRxiv*, 2019.

Reviewer #1 (Remarks to the Author):

This manuscript by Chung et al. described a procedure to enhance “2D clustering” of single particle cryo-EM particle images. The pre-process using a method “2SDR” that was described in a separate and published paper to reduce noise (denoise) of particle images, so that 2D classification of such denoised images produced better results. Overall, this is an interesting procedure, and could be potentially interesting and useful to cryo-EM practitioners. One specific result, reducing the time for ISAC 2D classification, would be interesting and useful to deal with heterogeneous dataset. ISAC is known to be very computational expensive. Using the pre-process procedure described here, it would make the ISAC more applicable. Overall, this is an interesting manuscript.

Q1: Authors demonstrated the improvement of 2D classification by using pre-processing procedure (2SDR) that was described in a previous publication. Reading from the manuscript, authors emphasized the importance of pre-processing by 2SDR, which was already published. Intuitively, I would imagine that any improvement of SNR of particle images, by whatever method, would lead to a better classification. If so, intuitively, the procedure described here can be simplified as to state that improving SNR of particle images helps better 2D classification. For example, deep learning convolutional neural network denoise could potentially also lead to the similar improvement. I would suggest author to provide clearer description if this is improvement of 2D classification is a result of improved SNR, or specifically by 2SDR? Regardless of the result, since 2SDR is already published, it would not reduce the potential interest of this manuscript.

Reply to Q1: We are very grateful for the encouraging and insightful comments. Intuitively, 2D classification would benefit from the direct usage of denoised particles due to the improved SNR. However, we found this is not the case when we tested this idea by directly using the denoised particles generated from 2SDR. Similar observations were reported on direct usage of denoised particles in 3D reconstruction has produced unreliable or worse results than raw particles (Bepler, *Nature Methods* 16, 1153). One obvious reason to explain this is that the loss of particle information due to denoising would hurt the performance of 2D classification. In this work, we found a scheme to couple the denoising with 2D classification that can avoid

information loss to produce meaningful results. This scheme, presented as the Pre-processor here, entails (1) perform reference-free alignment on denoised particles, (2) extract alignment parameters, (3) applied these parameters to original raw particle images, (4) feed the re-positioned raw particles to a 2D classification. As a result, the particles fed into a 2D classification algorithm have the same SNR as the original “non-denoised” particles. The improvement of 2D classification is likely from the improved particle alignment in the beginning through the improved particle SNR.

Our work on the pre-processor suggests one may use the same strategy to couple other denoising algorithms such as CNN or other learning methods such as TOPAZ to 2D classification --using the parameters extracted from the denoised set and apply them to the original set. Presumably, de-noised particles from CNN or other learning method like TOPAZ may give similar improvement. But whether or not this idea is true and how much an algorithm can achieve warrant further study. Importantly, compared to 2SDR de-noising that uses the data directly and very fast, CNN or TOPAZ needs training sets and the total computation should be more expensive.

Q2: It would be nice to have quantitative estimation of SNR improvement by the 2SDR pre-processing procedure. This could give a done by quantitating spectra SNR of original raw image verses images after 2SDR denoising procedure.

Reply to Q2: We now provide quantitative estimation of spectra SNR improvement by 2SDR. We use ASPIRE [1] from Amit Singer group to estimate the particle SNR, and it shows that it improves the SNR of 70 ribosome from 0.019 to 0.805 and that of TRPV1 from 0.003 to 2.717. It should be noted again that the particles fed into RELION or ISAC 2D classification are re-positioned “raw” particles, which have the same unimproved SNR as the raw particles.

	70s	Betagal	80s	TRPV1	NanoD
original	0.019	0.037	0.010	0.003	0.006
After 2SDR	0.805	1.665	3.317	2.717	2.850

*Q3:*The statement “confirmed by cryo-EM experts” is very subjective. Instead, the success of the procedure should be evaluated by quantitative

measurement. In this case, one possible parameter is to compare the total number particles within the “good” classes selected. If the number in (b) is more than that in (a), the procedure at least rescued more particles. While this is not entirely objective, it at least provides some “semi” quantitative measurement that is better than “confirmed by cryo-EM experts”.

Reply to Q3: We now remove this subjective statement. As for an objective criterion to select good classes, we can use the alignment statistics. Based on this criterion, the findings of increase in the number of good classes and the total number of particles are consistent with those by inspection.

Minor:

Q4: “Clustering” is a rather vague and confusing term. My understanding is that this is a 2D classification procedure. I suggest authors to change this to “2D classification”, which is more accepted term.

Reply to Q4: As suggested, we replace “2D clustering” with “2D classification” in the **title** and in the revised text.

Reference

[1] Algorithms for Single Particle Reconstruction (ASPIRE),
<http://spr.math.princeton.edu/>

Reviewer #2 (Remarks to the Author):

In this manuscript Chung et al. proposed a pre-processing step to facilitate 2D clustering in cryo-EM data processing. It is a PCA based algorithm which can de-noise particle images fast. Implemented into software package Relion or ISAC, it provided potential improvements in 3 aspects. First, it could make data processing faster. Several benchmark data sets were tested with, and showed improvement in some cases. Secondly, it could improve the quality of class-average images, which could be checked visually, or reflected by a better initial model generated from these images. Lastly, it could increase the yield of good particles, which improved the final resolution by 0.03 - 0.09 Å; with Relion and 0.1-0.2 Å with ISAC.

The proposed algorithm is novel in cryo-EM data processing and can be implemented easily. However, the major benefit of the pre-processor is to increase the speed of 2D clustering using ISAC. It doesn't show convincing improvements when used with Relion, which has solved the largest number of

reconstructions deposited in EMDB. It's unclear whether it will be helpful when using with other widely-used software, e.g., CryoSparc, which runs faster than Relion and ISAC, and produced reconstruction with the highest resolution to-date, or EMAN2, which has neural network features for better performance.

Rely: We thank the review for insightful review. We appreciate the review acknowledges that this pre-processor is novel and has provided benefits on 2D classification. It should be stressed the making of the pre-processor aims to mainly help 2D classification to do its job of particle curating better or faster. To this end, our works demonstrate that the pre-processor has fulfilled its purpose by showing that the pre-processor can increase the yield of particles, make 2D class faster, and increase the number of good classes. We agree that the impact of the pre-processor made on the final maps seems not to be convincing enough in particular for the tests on Relion. We would also like to explain that there is no warrantee that the pre-processor would significantly impact the 3D reconstruction because it depends on the subsequent processing steps in the workflow and also on the quality of the data. Nonetheless, in this revision we went on to test non-curated datasets of high heterogeneity as suggested by Reviewer2 and also by Reviewer3.

Fortunately, the results of testing on non-curated datasets show that the pre-processor confers larger impact on the final map for more heterogeneous data, highlighted by a TRPV1 embedded in nano-disc (>210,000 particles released to EMDB by Yifan Cheng) extended from 3.01 Å to 2.82 Å by the pre-processor applied on Relion 2D classification. Surprisingly, this result surpasses 2.95 Å in the original work that used both Relion 2D and 3D classifications. Such news would be very encouraging to the users of Relion, by which the largest number of cryo-EM structures in PDB has been solved.

In the wake of cryo-EM resolution revolution, there have been zealous pursuits in improving the resolution to suit pharmaceutical applications. In this light, the gain of 0.20 Å by the pre-processor is significant as it is comparable to that made by correction methods -- for example, by using MotionCorr2 (Zheng et al., Nat Methods 14: 331-332 (2017)), or Unblur (Grant and Grierieff (2015), ELife 2015;4:e06980). Compared to those correction methods, the pre-processor is really inexpensive and can be used together to have combined effect.

The gain of 0.20 Å in the resolution beyond 3.0 Å is very challenging --- given a B factor of about -100 \AA^2 , to increase the resolution by 0.20 Å requires increasing the number of particles at least 3 times (thanks to the plot given by

Rado, see Danev, R., Tegunov, D. & Baumeister, W. (2017), Elife 6, e23006). In this view, the pre-processor is valuable as it precludes the need of collecting more data. However, it should be pointed out that there would be no measurable improvement to be made for highly homogeneous data, as exemplified by the case of 80S ribosome.

As suggested, we also test the pre-processor with Relion3 and CryoSparc 2D classification to find similar and proportional benefits. As for the users of EMAN2 CNN or NN in other packages, we explain the user can use our offered plug-in to accommodate the particles denoised through the neural network to potentially have the similar benefits.

Queries:

Q1. There's little improvement of speed when using the pre-processor with Relion, as summarized in Table 1. The authors suggested that it was because Relion would run a fixed number of iterations based on the parameter given. Supplement Fig 4 shows that less iterations are needed with the pre-processor and thus save time. However, the claimed visual improvements between the two groups of figures are unclear.

Reply to Q1: We agree that visible improvements between the two groups of figures are unclear. We now use the initial models from different iterations for comparison. As shown in **Figure 1** below, the “with-the-preprocessor” model from 10 iterations is comparable to the “without the-preprocessor” model from 25 iterations.

Figure 1 Left column shows the results without the pre-processor and the right column the results with the pre-processor. From top to bottom: the number of iterations is 5,10,15,25, respectively. Good classes are framed or boxed in red.

2D classification

Initial model

Iteration

Q2. The pre-processor has been tested with Relion2. However, Relion3 published in 2018 is 1.5-2.5x faster than Relion2. Can the pre-processor be coupled with this newer version and if so, how much improvement in speed will it provide?

Reply to Q2: Yes – we test Relion3 to find the pre-processor can *also* be coupled to the Relion3 2D classification to give similar results. We also confirm 2D classification on Relion3 is 1.6 times faster than Relion2. The pre-processor does not improve the speed of Relion2 or Relion 3 when fixed

number of iterations is used. However, as shown in the reply to Q1, the number of iterations may be reduced with the aid of pre-processor, which is also true to Relion3. The improvement of speed is expected to be proportional.

Q3. Supplement Fig 5, which supports statistical improvement in Relion, is not clear. All the figures in column 1 and 3 shares the same X-axis. Each figure represents one experiment, and two data points were shown in each figure. The 2nd data points were X=999 in all figures, with explanation for the meaning of value 999. However, the X values for the 1st data point were not consistent, being 1, 3 or 4 in each figure. Similarly, figures in column 2 and 4 have X = 1, 2, 3 or 5. Why are these certain values chosen? When considering all data points for each experiment, does the statistics still show improvements?

Reply to Q3: As shown in **Table 1**, the X value in each experiment in the manuscript was a threshold chosen to be the smallest integer that is larger than the values of alignment accuracy. So, it varies. To avoid the possible confusion, we now only divided the results into groups with stable and non-stable alignment, where the stable ones are those whose alignment statistics is other than 999.

Table 1: Summary of alignment statistics of Relion3

	Relion 3	
	70S ribosome (with/without the pre-processing)	Beta-gal (with/without the pre-processing)
Number of good classes by alignment statistics	18/22	12/16
The rlnAccuracyRotations of good class (All others than 999)	(1.52,1.86,1.90,2.15,2.02,2.60,2.25,1.90,1.94, 2.60,2.15,1.70,2.60,3.80,2.10,3.05,3.60,3.10)/ (1.70,1.90,2.60,2.60,2.30,2.60,2.08,2.60,1.90,3.10,2.10,2.06, 2.60,1.90,2.60,2.60,2.35,2.60,3.80,3.80, 2.10,3.10)	(1.90,2.60,2.70,2.60,2.95,2.60,2.85,2.60,2.60,2.95,2.80,2.60)/ (1.70,1.76,1.90,1.94,1.96,2.08,2.10,2.30,2.40,2.55,2.60,2.60, 2.75,2.90,2.80,2.90)

Q4. The authors showed that their pre-processor reduced time spent on 2D clustering using ISAC by 30~40%. 2D clustering is one of the seven major steps for data processing. What percentage of improvement is achieved when considering the whole work flow?

Reply to Q4: Take the beta-galactosidase dataset for example, the pre-processor reduces the time of ISAC on this dataset by 28 % (see below), by which the time of the whole work flow is reduced by 22%. The measurements of each step are given in **Table 2** below. When working on large dataset, the fraction saved on 2D classification will be close to that saved

on the entire workflow due to other steps taking much less time.

Table 2: Computation time on Beta-galactosidase dataset

	ISAC/P-ISAC (Hour)
Movie Alignment	0.19
CTF Estimation	0.02
Particle Picking	0.10
2D classification	1.58/1.14
Initial Model	0.02
3D Refinement	0.10

Q5. Are the data used here damage-compensated or not? Damage compensation also de-noises particle images, which could have a similar improvement as the pre-processor. It would be helpful for the readers to know whether the pre-processor work with damage-compensated data.

Reply to Q5: The data tested in this study have been compensated with radiation issue.

- (i) 80S ribosome (EMPIAR-10028, curated):
Damage compensation by B-factor weighting, the pre-processor still improves the final map by 0.1 Å.
- (ii) TRPV1 (EMPIAR-10005, non-curved set):
Damage compensation by using dose-limited data in the end, the pre-processor shows improvement of 0.2 Angstrom for both Relion2 and for ISAC.
- (iii) TRPV1 in nanodisc (EMPIAR-10059, non-curved set):
Damage compensated by dose weighting, the pre-processor shows improvement of ~0.2 Angstrom for Relion3.

Taken together, these tests demonstrate that the pre-processor can further work on damage-compensated data. However, the scale of improvement depends on many other factors. Radiation damage occurs as electron dose accumulates on the specimen and its degree is dependent on spatial frequency. Damage compensation becomes possible with movie camera allowing for dose fractionation, by which high-resolution information damaged in later frames can be down-weighted. As a result, “de-noising like” enhancement can occur due to the boost of low-resolution information as the aligned frames are summed. This issue has been best addressed by Tim

Grant and Niko Grigorieff in *eLife* 2015, 4:e06980 at the time of resolution revolution; they implemented it with the *Unblur* package that combines frame alignment and a nice “Exposure Filtering” algorithm for compensating on over-dosed data. It is noted that alternative approaches such as B-factor weighting, dose-limiting and dose weighting give similar results as *Unblur* as shown in the MotionCor2 paper by Zheng, Yifan Cheng and David Agard. This means our pre-processor (particle filter) should further work with “*Unblur-ed*” data. It should be noted that the SNR boosted by 2SDR is quite significant.

	70s	betagal	80s	TRPV1	NanoD
original	0.019	0.037	0.010	0.003	0.006
After 2SDR	0.805	1.665	3.317	2.717	2.850

Q6. Fig 2e, 2f, 3e and 3f show initial models were improved. The authors suggested that they were proof of higher quality of class averages thanks to the pre-processor. However, quality of any model is also based on numbers of particles or class averages, which were not consistent in these examples. Therefore, these models show that the pre-processor can help with the generation of initial model, which is beneficial for the whole work flow, but not necessary to be related to better class averages.

Reply to Q6: We agree –we cannot ascribe the better initial model to higher quality of class averages since there is a confounding factor of increased number of good class averages. We re-phrase it as suggested.

Q7. Improved local resolution shown in Fig 4h is hard to see as most parts of the figures are in blue color. Scaling the coloring differently or zooming in to the desired region could be helpful.

Reply to Q7: We now re-scale the color in “Figure 4h” such that the more flexible parts are clearly in red in the density maps. Side-by-side comparison indicates some red regions are reduced through the pre-processing. We also zoom in the drug emetine as shown below.

Q8. Reported resolution improvement is 0.03 - 0.2 Å, which are not significant. This could be that the two small data sets are limited by number of particle, and the 80s ribosome data had reached the highest possible resolution determined by Nyquist frequency. Maybe different data should be tested to show the potential improvement from filtering.

Reply to Q8: Improving resolution from 3.5 Å on is difficult and the difficulties increase as one moves into the sub 3Å regime -- for example the beta-gal case (Bartesaghi et al., Structure 26, 848 (2018)), correction methods lead only to marginal improvement --0.07 Å made by local defocus correction, 0.09 Å by per particle motion, and 0.12 Å by dose weighting; however, they together add up to 0.4 Å. We have discussed the importance of these efforts in the wake of cryo-EM resolution revolution.

Nonetheless, since another reviewer also made suggestions to test more heterogeneous datasets such as non-curated data and channel in nanodisc, we thus tested the non-curated set of TRPV1 (EMPIAR-10005) that contains ~80,000 particles originally solved to 3.4 Å (0.7 Nyquist), and TRPV1 embedded in nanodisc (10059) that contains ~220,000 particles originally to 2.95 Å (0.83 Nyquist).

As listed in Table 3, the improvement on non-curated TRPV1 is from 3.57 to 3.42 Å by one-pass of the pre-processing applied to Relion and the second pass further improves it to 3.37 Å. On the nanodisc-embedded TRPV1, it is improved from 3.01 to 2.86 Å by one-pass while another pass extends it to 2.82 Å (Table 4). Such advance in resolutions is associated with evident improvement of map quality --- as shown by the maps of TRPV1 in nanodisc below, the map is evidently improved in the lower part that contains the ankyrin repeats.

Table 3: Summary of one-pass results on **EMPIAR-10005**
(Relion (with 200 Classes) and ISAC (Bin 3X and 2000 per class))

	Relion	P-Relion	ISAC	P-ISAC
Number of Classes by inspection	15	20	25	27
Number of Classes by alignment statistics	25	31	NA	NA
Particles	50,620	55,269	43,690	52,661
Resolution (Å)	3.57	3.42	3.56	3.39
Time(Hour)	13.56	13.52	10.56	8.48

Table 4: Summary of one-pass results on **EMPIAR-10059** (100 Classes)

	Relion	P-Relion
Selected number of classes by inspection	12	13
by alignment statistics	20	23
Particles	153,839	166,236
Resolution (Å)	3.01	2.86
Time (Hour)	9.49	9.55

Reviewer #3 (Remarks to the Author):

The manuscript by Chung et al. is well written and easy to follow. It proposes a modified workflow for processing single particle cryo-EM data which enhances 2D clustering of particles and improves the resolution of reconstructed volumes for a variety of test datasets. This is achieved by adding a

preprocessing step that uses a denoising algorithm (2SDR). All tested cases benefit from the preprocessing step with a minimal added computational cost. Also, the generation of the initial reference model is improved with the use of preprocessed data.

My main concerns are:

2D clustering is mainly used to clean Cryo-EM datasets and separate real particles from non-particles or contaminations. None of your 2D classes showed like contamination features. This seems normal given the nature of curated test datasets, but how the preprocessing step will perform in the presence of increased heterogeneity and contaminations?

Although all tested cases benefited from the preprocessing step, it seems from your results that *the number of classes* in Relion or down-sampling in ISAC have a bigger impact on the results *than* the use of the proposed additional preprocessing step.

Given this, 2D clustering with *optimized parameters such as the number of classes, regularization parameter T , angular sampling* for Relion may not benefit as much from the preprocessing step. In some conditions, for the tested datasets, improvement is marginal.

Do you expect that datasets/particles closer to focus would be the ones that benefit more of the preprocessing step?

These and other comments are better explained in the attached review file and need to be addressed.

Thank you for the stimulating manuscript.

The manuscript by Chung et al. proposes the use of a preprocessor (2SDR; denoising algorithm) as an additional step for the common workflow for processing cryo-EM SPA data to enhance 2D clustering with a relatively minimal added computational cost. The use of the preprocessor increased the number of well-defined clustered classes and with this the number of selected particles, producing maps of higher resolution and in some cases an improvement of the map interpretability and agreement of the maps with the models.

The manuscript is well written and easy to follow, it includes proper background of the basics and the general workflow of cryo-EM data processing.

Reply: We are very grateful for the insightful comments and valuable suggestions. We give a point-by-point reply as follows.

Comments

Q1. Number of classes in Relion or down-sampling in ISAC seem to have bigger impact on the results than the use of the proposed additional preprocessing step (*Table 1*). Given the influence of the number of classes in Relion, how would you expect changing parameters such as regularization parameter T and angular sampling will play with the proposed preprocessing, are they still going to benefit in the same proportion/amount.

Reply to Q1: Relion conforms to the Bayesian principle that one way to optimize the classification result is through tuning the relative weight of the experimental data and that of the prior, which can be done on Relion by using a regularization parameter T. Here, we perform three Relion tests on the 70S ribosome dataset with T=2, 4 and 8. Indeed, increasing T increases the number of classes in Relion.

The numbers of classes for Relion and P-Relion obtained with various T are summarized in a **Table 1** below. As shown in a **Figure 1** below, for every given T, the pre-processing consistently increases the number of good classes with similar quality. From this result, we expect that our pre-processing can benefit 2D clustering results for various T values. As for the angular sampling, our test indicates that changing it does not evidently change the number of classes as changing T does (not shown).

Table 1: Summary of alignment statistics under different T

Regularization	Number of classes based on alignment statistics	
	Relion	P-Relion
T		
2	18	22
4	23	26
8	23	29

Figure 1: Left column: without the preprocessor; right column: with.
(a) T=2 (Default).

(b) T=4

(c) T=8

Q2. Analyzed datasets comprise diverse test cases but is still remain an interesting question of how the preprocessing workflow will affect in cases of high heterogeneity or embedded particles in other structures such as nanodiscs ?

Reply to Q2: To test how the pre-processing will affect highly heterogeneous system such as non-curated data or particles embedded in nanodisc, we work on two datasets.

- (1) We visit the non-curated set of all particles of TRPV1 (EMPIAR-10005). The 871 summed micrographs in the repository have a total of 80443 particles. The test results on this non-curated data with Relion and ISAC (**Table 2**) both show that the pre-processing can improve the final resolutions while the improvements are larger than those on the curated data. Observing bi-particle contaminants appear in the class averages, we applied two runs of pre-processing. One run of 2D Relion classification with the pre-processor improve the resolution from 3.57 to 3.42 Angstrom (**Table 2**). Another run of P-Relion has extended the resolution from 3.42 to 3.37 Angstrom (**Table 2**), slightly better than that in the original work

obtained by 2D and 3D classifications.

Figure 2: Left panel without versus right panel with 2SDR-Preprocessor.

(a) Relion

(b) ISAC

(c) Second Run of Relion

Table 2: Summary of results on EMPIAR-10005
(Relion (with 200 Classes) and ISAC (Bin 3X and 2000 per class))

	Relion	P-Relion	ISAC	P-ISAC
Number of Classes by inspection	15	20	25	27
Number of Classes by alignment statistics	25	31	NA	NA
Particles	50620	55269	43690	52661
Resolution (Å)	3.57	3.42	3.56	3.39
Time(Hour)	13.56	13.52	10.56	8.48
	2 nd Relion	2 nd P-Relion		
Particles	39570	42868		
Resolution (Å)	3.43	3.37		
Time(Hour)	6.95	7.52		

(2) For nanodisc data, we worked on a TRPV channel in nanodisc (EMPIAR-10059) that contains ~220,000 particles. To speed up 2D classification, we used Relion3, which is 1.6X faster than Relion2, to test with and without pre-processing on. The test results show that the pre-processing gives similar benefits as it did for (1), as shown in **Figure 3A** and **Table 3** below. With pre-processing, we selected 166236 particles to get 2.86 Å. Without pre-processing we selected 153839 particles to get 3.01 Å. The comparison shows the pre-processing does again improve the final 3D resolution. Another run of P-Relion extends the resolution to 2.82 Å

(Figure 3B). By contrast, 73,929 particles were finally selected using 2D and 3D classifications by the authors in Yifan lab to obtain 2.95 Å resolution (see Ref [6]).

Figure 3A: Left column shows the results without the pre-processor while right column shows results from the pre-processor.

Table 3: Summary of results on EMPIAR-10059 (100 Classes)

	Relion	P-Relion	P-Relion (second
Selected class number	12	13	11
Alignment statistics	20	23	N/A
Particles	153839	166236	147749
Resolution (Å)	3.01	2.86	2.82
Time(Hour)	9.49	9.55	6.23

Figure 3B

Q3. The main common purpose of 2D clustering is to classify and clean particles from non-particles. Heterogeneity and damaged particles are commonly treated with 3D classification, but not only. In your case none of the classes show strong features of typical contaminations, current test datasets were testing clustering mainly with good particles. This makes sense that given the nature of the datasets they present little of no contaminations since they were curated and some are used for Relion tutorial. Do you expect a similar improvement on classification with contaminations?

Reply to Q3: As shown in the reply to Q2, classifying more heterogeneous datasets, including the non-curated set of all particles of TRPV1 (EMPIAR-10005) and a TRPV channel in nanodisc (EMPIAR-10059), we see larger improvements.

Q4. 2SDR denoising algorithm represents an essential part of the preprocessing proposed workflow, but in the current state of the manuscript it is not clear what makes 2SDR denoising a better option to denoise particles for clustering classification over other algorithms such as CWF[2] and neural network denoising algorithms which have been recently used with great results for particle picking in packages such as Warp[4], Topaz[1] and Cryolo/Janni[5, 3].

Reply to Q4: Compared to other de-noising algorithms, 2SDR de-noising has the strength in its low computation cost. In a separate manuscript we uploaded to arXiv, we have explained the cost of 2SDR is much lower than that of PCA due to the computation complexity being greatly reduced by a two-stage scheme. CWF [2] de-noises particles using steerable PCA and thereby has high computation complexity of (nL^3+L^4) [2] similar to that of PCA (L denotes the dimension of the image and n the number of images). As for those neural network methods, they require effective training datasets. It is well known that, to solve a novel structure problem, choosing an appropriate training dataset alone usually poses a challenging task. Finally, our pre-processing strategy avoids direct usage of de-noised particles in 2D classification because reliable results cannot be obtained using de-noised particles, either from our 2SDR or from neural network, for which we have tested Janni and others. The results of 2SDR and Janni are shown below (**Figure 4**). In the revision, we do discuss the possibility of accommodating neural network de-noised particles into the pre-processor using the plug-in we offer.

Figure 4 Left column: 2SDR de-noised particles and the Relion 2D classification results; right column: Janni de-noised particles and the Relion 2D classification results.

Q5. Starting references, generated with PRIME, use its own generated reference for each 3D refinement or the same reference for all cases? what is the proportional contribution of starting with a better reference? if any, or in other words does the quality of the initial reference affects results on top of the clustering? (particularly for poorly generated references). Was PRIME selected for any particular reason? In a recent article of Topaz-denoising[1], it is mentioned that for their case using denoised particles with Cryosparc 2 for Ab-initio model generation, produced less reliable results than using the raw particles. In your case, will results of Ab-initio model generation vary with different packages?

Reply to Q5:

- (1) PRIME was selected because we tested many to find it generates robust initial model from class averages. Importantly, we used the initial model produced from class averages, not from particles, to gauge the quality of the class averages.
- (2) Each case uses its corresponding initial model. Better reference model seems to have better final result ---this is more obvious for Relion because the curated particles from Relion and P-Relion are quite similar in several cases but the final 3D structures are quite different ---this implies the role played by the initial model.
- (3) Although we do not use the initial model generated from the particles in our workflow, we did try to check on initial models from the particles using CryoSparc---using raw or denoised particles. Here is what we found --- the initial model from the 2SDR denoised particles is not as reliable as that from the raw particles (**Figure 5**). This phenomenon is similar to what was found for Topaz de-noised particles [1].

Figure 5: CryoSparc generated initial models of 80S ribosome: from 2SDR de-noised particles (left panel) compared that from raw particles (right panel)

Q6. Do you expect that datasets/particles closer to focus would be the ones that benefit more of the preprocessing step? Can you comment on this with your current test cases.

Reply to Q6: This is truly an insightful question and it is reasonable to expect so since 2SDR de-noising improves image SNR. We then test this idea by using real data of 80S ribosome and simulation data from 70S ribosome for estimating the statistics.

- (1) For simulation data, we set the SNR the same (0.05) and create two 70S particle sets, one with defocus (1-1.5 μm) and the other with defocus (1.5 to 2 μm). The true positives found for without and with 2SDR for the lower defocus set are 51.3% and 93.7%. The true positives found for without and with 2SDR for the higher defocus set increase to 61.5% and 94.4%. The comparison shows lower defocus set would benefit more.
- (2) For the test on real experimental data, we first work on the 80S dataset since it was collected with a broad defocus range 0.8-3.8 micron. By using a medium defocus of 2 micron, we split the data into two halves. We then employ ISAC to do 2D classification. Interestingly, as shown in **Table 4** -- for the lower defocus subset, the final resolution is improved from 3.20 to 3.19 Angstrom with the pre-processor, while for the higher defocus subset the resolution remains unchanged at 3.3 Angstrom. Although the figure of

improvement is marginal, it shows a sign that larger benefit will be seen on closer-to-focus data. However, we refrain from emphasizing this point in the revised manuscript as we consider more extensive tests on closer-to-focus data are worth of further study.

Table 4: Classification results of 80S ribosome with high defocus set and low defocus set

(3) We further test 70S particles and plot the histograms of harvested particles in terms of defocus values (**Figure 6**).

Figure 6: Histogram of particle's defocus values in Angstrom ($10^{-4} \mu\text{m}$): number of particles versus defocus values. (a) Original 70S ribosome dataset. (b) Curated subset after performing ISAC. (c) Curated subset after performing P-ISAC.

Q7. It is mentioned in the manuscript that with the preprocessing step, convergence of 2D clustering is faster. What criteria or parameters were used to monitor convergence of clustering? When you mention you used up to 15 cycles for original data and 10 for preprocessed, had this already converged? If not, will giving proper time for both to converge, would this close the gap between the results?

Reply to Q7: We do not find using alignment statistics as the criteria can support this claim of convergence. We turn to check the initial model resulting from different Relion iterations shown in **Figure 7** below. It is evident that the pre-processing enables Relion to get reasonable results with less number of iterations. For example, 10 iterations with P-Relion is comparable to 25 iterations with Relion.

Figure 7A: Left column shows Relion 2D results while right column shows the corresponding results with 2SDR-Preprocessor. From top to bottom: the number of iteration is 5,10,15,25, respectively.

Figure 7B: Left column shows initial model results using original workflow

while right column the results from that with 2SDR-Preprocessor. From top to bottom are iteration 5,10,15,25, respectively.

Table 5: Summary of results from different iterations (without/with the Preprocessor)

Iteration	Particles	No. of manually selected class	Selected by alignment statistics
5	3196/3342	4/6	3/5
10	3818/4149	7/11	10/13
15	3904/4247	8/12	13/15
25	4146/4471	9/13	17/20

Q8. Figure 1a does not contribute for the purpose of explaining the modified workflow with the added preprocess step, preprocessing step could be added as a branch in a different color, to help the reader identify where it takes place and how its different from the classical workflow. This will also help to link the workflow with figure 1b. Some steps are missing arrows for possible looping steps.

Reply to Q8:

Q9. Figure 1b may give the sensation that a 3D refinement is performed on denoised particles and then substituted by raw particles for another 3D refinement, but if I understood correctly the first reference-free alignment is a 2D and thus it needs explicit specification where a 2D or a 3D reference-free alignment is performed. What is the contribution of extracting alignment

parameters and transfer them to the original images (are they really applied or only transferred to the appropriate files. *Are the 2D alignment parameters used to re-extract re-centered images, if so, this should be better explained*). If alignment parameters are not transferred to the original images, but only selection based on clustering is made, how much this affects currently reported results.

Reply to Q9:

- (1) We now specify the reference-free alignment as “2D reference-free alignment”.
- (2) The extracted alignment parameters are applied to raw particle images -- they are re-centered and rotated. The classification algorithm then works on the re-centered/rotated raw particles, not the de-noised ones.

Q10. Figure 5 would benefit from a better labeling of graphs and volumes. In caption of image, for e) f) it is mentioned a map to model FSC but it seems that only g) shows FSC curves. It needs better explanation or rephrase. For i) what is the threshold level for displayed volumes? is the same for all of them?

Reply to Q10: We rephrase as follows.

Figure 5:(e) the statistics of the total number of harvested particles. Six clustering experiments under different settings are conducted (without and with the pre-processor colored with green/blue). 1: RELION/P-RELION with 175 classes, 2: RELION/P-RELION with 100 classes, 3: RELION/P-RELION with 50 classes, 4: ISAC without down-sampling, 5: ISAC/P-ISAC with 2X down-sampling, 6: ISAC/P-ISAC with 3X down-sampling. (f) the overall final resolutions corresponding to those in (e). (g) the map-to-model FSC curves by comparing the maps of RELION of 1, P-RELION of 1, ISAC of 6 and P-ISAC of 6 in (e) to PDB 3j5p. The calculation was performed using Phenix [29]. (h) the overall density maps of RELION of 1, P-RELION of 1, ISAC of 6 and P-ISAC of 6 in (e). The beta-sheets and the ankyrin repeats are indicated with a red arrow and a blue arrow, respectively. The B-factors measured and used for sharpening the 4 maps are -86.1,-90.1,-90.0 and -92.7, respectively. (i) Enlarged densities corresponding to (h) where the same threshold of 0.49 on Chimera was applied. The left helix corresponds to 559-628 of subunit C, displayed as a reference. The middle and the right helices correspond to 429-455 and 469-498 of subunit A and they form a "U" with the connecting loop (456-468). The brown arrow denotes 464-468 while the green arrow denotes 456-458.

Q11. All figures could benefit from a better indication/grouping in the figure to recognize quickly which classes and volumes come from original data, preprocessed data, Relion or ISAC.

Reply to Q11: We will use this following style that has annotation on it.

References

- [1] Tristan Bepler, Alex J. Noble, and Bonnie Berger. Topaz-denoise: general deep denoising models for cryoem. bioRxiv, 2019.
- [2] Tejal Bhamre, Teng Zhang, and Amit Singer. Denoising and covariance estimation of single particle cryo-em images. J Struct Biol, 195(1):72–81, 07 2016.
- [3] Jaakko Lehtinen, Jacob Munkberg, Jon Hasselgren, Samuli Laine, Tero Karras, Miika Aittala, and Timo Aila. Noise2noise: Learning image restoration without clean data, 2018.
- [4] Dimitry Tegunov and Patrick Cramer. Real-time cryo-electron microscopy data preprocessing with warp. Nat Methods, 16(11):1146–1152, 11 2019.
- [5] Thorsten Wagner, Felipe Merino, Markus Stabrin, Toshio Moriya, Claudia Antoni, Amir Apelbaum, Philine Hagel, Oleg Sitsel, Tobias Raisch, Daniel Prumbaum, Dennis Quentin, Daniel Roderer, Sebastian Tacke, Birte Siebolds, Evelyn Schubert, Tanvir R. Shaikh, Pascal Lill, Christos Gatsogiannis, and Stefan Raunser. Sphire-cryolo: A fast and accurate fully automated particle picker for cryo-em. bioRxiv, 2019.

- [6] Gao Y, Cao E, Julius D, and Cheng Y. TRPV1 structures in nanodiscs reveal mechanisms of ligand and lipid action. Nature 534:347–351. 2016.

REVIEWERS' COMMENTS:

Reviewer #2 (Remarks to the Author):

The authors have addressed all my concerns in their revised manuscript. The improvement by the procedure on 2D classification is more clearly demonstrated by the new data added.

Reviewer #3 (Remarks to the Author):

Thank you for the corrections in the revised manuscript by Chung et al. All the previously raised questions and comments have been satisfactorily addressed. Modifications on the revised manuscript have improved readability and clarity.

The manuscript describes an optional added step to the common processing workflow for cryo-EM single particle analysis (SPA). The use of a denoising preprocessing step improves 2D classification and provides a modest enhancement in the resolution and interpretability of the final maps of tested cases. It also has the added benefit of speeding up 2D classification while not adding a large amount of extra computation time. The current manuscript constitutes a good example of the possibilities and available room for improvement on the workflow for cryo-EM SPA and opens the possibility for further exploration and improvement with the use and coupling of other steps, different software, and denoising algorithms.

Minor comments:

The manuscript mentions an "offered plug-in" two times in the discussion section and one in the code availability section. This can give the impression of a software implementation but no reference to where to obtain it, is provided. Otherwise, it should be stated that the method constitutes a plug-in to the common workflow rather than a provided plug-in.

It is stated in the discussion section "we used CryoSparc since it has produced reconstructions with the highest resolution to date". It is not clear if you mean it has produced the highest resolution to date for your case or in general. If the later, it may not be true, as recently reported structures with resolutions $<2\text{\AA}$ use different software. Along the same lines, you mention "the pre-processor can further improve the final resolutions to unprecedented level -3.10\AA for 80S ...", it can give the impression that 3.1\AA represents, from all available structures, an unprecedented resolution for 80S which is not true. Is the minus sign of -3.10\AA a typo? (there are several "-" in MS that should be checked) I was not able to

find if refinement in Cryosparc was non-uniform refinement or only homogeneous refinement.

Hyperlink to EMPIAR-10059 is broken.

Reviewer#3

Minor comments:

The manuscript mentions an “offered plug-in” two times in the discussion section and one in the code availability section. This can give the impression of a software implementation but no reference to where to obtain it, is provided. Otherwise, it should be stated that the method constitutes a plug-in to the common workflow rather than a provided plug-in.

Reply: We now state in Discussion “To facilitate the usage of the preprocessor, we provide a plug-in that permits its insertion between particle picking and 2D classification (Figure 1(b))”. We offer the code of the plug-in with a statement in Code Availability “The request can be made by applying through <http://sabid.stat.sinica.edu.tw/>.”

It is stated in the discussion section “we used CryoSparc since it has produced reconstructions with the highest resolution to date”. It is not clear if you mean it has produced the highest resolution to date for your case or in general. If the later, it may not be true, as recently reported structures with resolutions $<2\text{\AA}$ use different software. Along the same lines, you mention “the pre-processor can further improve the final resolutions to unprecedented level -3.10\AA for 80S ...”, it can give the impression that 3.1\AA represents, from all available structures, an unprecedented resolution for 80S which is not true. Is the minus sign of -3.10\AA a typo? (there are several “-“ in MS that should be checked) I was not able to find if refinement in Cryosparc was non-uniform refinement or only homogeneous refinement.

Reply:

- (1) As for CryoSparc, we now change our wording to “In the step of 3D refinement, we used CryoSparc homogeneous refinement for its rapidity..”
- (2) As for the issue of final resolutions, we now change our wording to “the pre-processor can further improve the final resolutions to surpass the previous work: 3.10\AA for the malaria 80S ribosome, 3.20\AA for the TRPV1, and 2.82\AA for the TRPV1 In nanodisc.
- (3) As for the symbol of hyphen, we now take care of it to remove the confusion.

Hyperlink to EMPIAR-10059 is broken.

Reply: We have fixed all the links of 10059.